

# Identification of key parameters controlling demographically structured vegetation dynamics in a Land Surface Model [CLM4.5(ED)]

Elias C. Massoud[1,2], Chonggang Xu[3*], Rosie Fisher[4], Ryan Knox[5], Anthony Walker[6], Shawn Serbin[7], Bradley Christoffersen[8], Jennifer Holm[5], Lara Kueppers[5], Daniel M. Ricciuto[6], Liang Wei[3], Daniel Johnson[3], Jeff Chambers[5], Charlie Koven[5], Nate McDowell[9], and Jasper A. Vrugt[2,10]

[1]Jet Propulsion Laboratory, California Institute of Technology, Pasadena, CA, USA
[2]Department of Civil and Environmental Engineering, University of California Irvine, Irvine, CA, USA
[3]Earth and Environmental Sciences Division, Los Alamos National Laboratory, Los Alamos, NM, USA
[4]Terrestrial Sciences Section, National Center for Atmospheric Research, Boulder, CO, USA
[5]Climate and Ecosystems Devision, Lawrence Berkeley National Laboratory, Berkeley, CA, USA
[6]Environmental Sciences Division, Oak Ridge National Laboratory, Oak Ridge, TN, USA
[7]Environmental & Climate Sciences Department, Brokenhaven National Laboratory, Upton, NY, USA
[8]Department of Biology, University of Texas Rio Grande Valley, Edinburg, TX, USA
[9]Pacific Northwest National Laboratory, Richland, WA, USA
[10]Department of Earth System Science, University of California Irvine, Irvine, CA, USA

**Correspondence:** C. Xu (cxu@lanl.gov)

**Abstract.** Vegetation plays a key role in regulating global carbon cycles and is a key component of the Earth System Models (ESMs) aimed to project Earth's future climates. In the last decade, the vegetation component within ESMs has witnessed great progresses from simple 'big-leaf' approaches to demographically-structured approaches, which has a better representation of plant size, canopy structure, and disturbances. The demographically-structured vegetation models are typically controlled by a

5   large number of parameters, and sensitivity analysis is generally needed to quantify the impact of each parameter on the model outputs for a better understanding of model behaviors. In this study, we use the Fourier Amplitude Sensitivity Test (FAST) to diagnose the Community Land Model coupled to the Ecosystem Demography Model, or CLM4.5(ED). We investigate the first and second order sensitivities of the model parameters to outputs that represent simulated growth and mortality as well as carbon fluxes and stocks. While the photosynthetic capacity parameter ($V_{c,max25}$) is found to be important for simulated carbon

10   stocks and fluxes, we also show the importance of carbon storage and allometry parameters, which are shown here to determine vegetation demography and carbon stocks through their impacts on survival and growth strategies. The results of this study highlights the importance of understanding the dynamics of the next generation of demographically-enabled vegetation models within ESMs toward improved model parameterization and model structure for better model fidelity.

## 1   Introduction

15   Earth System Models are abstract representations of nature used to simulate physical, chemical, and biological processes across the interacting domains of the Earth system to estimate past, present, and future climate (*Claussen et al.*, 2002; *Dunne et al.*,





2012; *Arora et al.*, 2013; *Hurrell et al.*, 2013). The land component of Earth System Models, land surface models (LSMs), are capable of representing vegetation dynamics through the use of dynamic global vegetation models (*Foley et al.*, 1996; *Cox et al.*, 2000; *Krinner et al.*, 2005; *Friedlingstein et al.*, 2006; *Sato et al.*, 2007; *Arora et al.*, 2013). While first generation dynamic vegetation models represent plant communities and their competition using a single area-averaged representation

of each plant functional type (PFT) for each climatic grid cell (*Cox et al.*, 2000; *Pan et al.*, 2002; *Hickler et al.*, 2004), second-generation vegetation demographic models have emerged to better capture coexistence and competition driven by light-competition between different sizes of trees in a vertical canopy structure as well as successional dynamics through the representation of disturbance history (*Moorcroft et al.*, 2001; *Thonicke et al.*, 2001; *Sitch et al.*, 2003; *Hickler et al.*, 2004; *Fisher et al.*, 2010; *Scheiter et al.*, 2013; *Fisher et al.*, 2018). These next generation models allow comparison with many more

observed vegetation processes than first generation models, but also contain more degrees of freedom causing them to be more complex and subject to high variability.

LSMs typically contain a suite of different parameters to resolve the carbon, water, and energy fluxes and pools at the land-atmosphere interface (*Noilhan and Planton*, 1989; *Bastidas et al.*, 1999; *Gupta et al.*, 1999; *Masson et al.*, 2003; *Sargsyan et al.*, 2014). Many of these parameters can be estimated directly in the field, but others are difficult or impossible to measure

due to various complications such as the lack of a physical meaning, technological limitations, or spatial/temporal aggregation (*Entekhabi and Eagleson*, 1989; *Kumar et al.*, 2006). Parameters that are observable in the field are also often subject to large natural variability, including changes through space and time (*Wood et al.*, 1992; *Masson et al.*, 2003; *Fisher et al.*, 2015). For example vegetation parameters can be used to describe different root profiles (*Vrugt et al.*, 2001; *Zeng*, 2001) or photosynthetic capacities (*Leuning*, 2002; *Rogers*, 2014), however, model parameter values are often taken from literature

publications or databases, and may not represent local variation or capture seasonal or ontogenetic changes. For parameters of critical importance, even a small difference in parameter values can lead to significant divergence for multi-model ensemble projections or uncertainty in model predictions from different models (*Sitch et al.*, 2008; *Dietze et al.*, 2014; *McDowell et al.*, 2016; *Rogers et al.*, 2017). Since parameters are often defined in simulations with limited prior knowledge of their mean values and variation (*O'Hagan and Leonard*, 1976; *Kitanidis*, 1986; *Geromel*, 1999), a model sensitivity analysis is typically required

to adequately quantify the impact of the parameters on the model outputs for a better understanding of model dynamics and identification of targeted parameters for model calibration. Despite the need for such studies, systematic investigation of the parameter sensitivity of LSMs is not standard practice, potentially on account of the high dimensionality involved (although c.f. *Zaehle et al.* (2005); *Fisher et al.* (2010); *Pappas et al.* (2013)).

Today, many sensitivity analysis techniques are available (*Sobol'*, 1990; *Helton*, 1993; *Saltelli et al.*, 2000; *Razavi and Gupta*,

2016). Some of these methods examine the response of the outputs by varying input parameters one at a time and holding other parameters at their default values (*Saltelli et al.*, 2000). However, the sensitivity index derived by this type of assessment depends on the default values of the other parameters, and the assumption these values are satisfactory is questionable (e.g. *Da Rocha et al.* (1996); *Sen et al.* (2001); *Groenendijk et al.* (2011); *Schwalm et al.* (2010)) since the discrepancies in LSM predictions are strongly tied (through feedbacks of momentum, energy, mass and biogeochemistry) to the differences in their

representation of the land surface (*Crossley et al.*, 2000; *Rosolem et al.*, 2013). Therefore, it is desirable to use more extensive

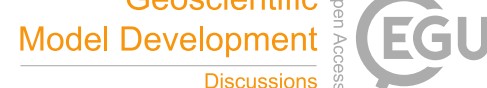

sensitivity analysis techniques that examine the response of model outputs averaged over the variation of all the parameters. These 'global' methods are generally preferred when computing power is not a limiting factor. In this study, we apply a 'global' sensitivity analysis to determine the influential parameters over a specified region of the parameter space. So far, there are several uncertainty and sensitivity analyses being conducted for size-structured land surface models (*Pappas et al.*, 2013;

*LeBauer et al.*, 2013; *Wang et al.*, 2013; *Dietze et al.*, 2014). To the best of our knowledge, this paper presents one of few formal global sensitivity analyses for an LSM with cohort-based vegetation demography (see *Pappas et al.* (2013)) and thus could provide important knowledge for model calibration and understanding of the model structure.

Our goal for this study is to apply a global sensitivity analysis, the Fourier Amplitude Sensitivity Test (FAST), to the CLM4.5(ED), at a tropical site to 1) understand the baseline model behaviors of vegetation carbon stocks and fluxes and

vegetation demography in relation to different model parameters and 2) provide directions for improved model parameterization toward a better model fitting to observations. Specifically, we aim to answer the following question: what are the main parameter controls on vegetation processes such as growth and mortality and on the resulting dynamics of carbon fluxes and stocks? Based on our understanding of simulated processes in CLM4.5(ED), we propose to test three hypotheses. Our first hypothesis is related to photosynthetic capacity. The carbon input for vegetation growth is through photosynthesis and in most LSMs, it

is simulated based on the Farquhar model (*Farquhar*, 1989) with the photosynthetic capacity represented by the maximum carboxylation rate at 25 ° C ($V_{c,max25}$) and maximum electron transport rate at 25 ° C ($J_{max25}$). $J_{max25}$ is commonly simulated in proportion to $V_{c,max25}$ in many models and previous sensitivity analysis studies (*Pappas et al.*, 2013; *Sargsyan et al.*, 2014; *Dietze et al.*, 2014) have shown that $V_{c,max25}$ is generally an important parameter that affect simulated carbon fluxes. Therefore, we hypothesize that the photosynthetic capacity parameter, $V_{c,max25}$, is a key control on simulated carbon fluxes in CLM4.5(ED)

(H1). Second, for demographic models, the allometry of trees determines the amount of carbon input to different tissues (e.g, leaf, root and stem). If the carbon is allocated more to leaf compared to stem, the tree will have a higher productivity but can also lead to lower stem growth and thus less height growth for light competition. Therefore, we hypothesize that allmoetry parameters are important for vegetation growth and long-term carton stocks (H2), as they will determine plant's growth strategies. Finally, the carbon stock for vegetation is affected not only by the input of carbon through photosynthesis,

but also by the loss of carbon through mortality. Therefore, we hypothesize that the parameters determining the mortality are important drivers of the long-term vegetation carbon stocks (H3), as they will control the length of carbon turnover time.

## 2 Materials and methods

### 2.1 CLM4.5(ED)

CLM4.5(ED) is an open-source land surface model coupled with a demographically structured dynamic vegetation model

to predict climate-vegetation interactions. The CLM is used within various Earth system modeling frameworks, including the Community Earth System Model (CESM) and the Norwegian Earth System Model (NorESM) (*Lawrence et al.*, 2011; *Bonan et al.*, 2011). The Ecosystem Demography Model (ED) is a dynamic vegetation model that scales up the behavior of forest ecosystems by aggregating individual trees into representative 'cohorts' based on their size and PFT, and by aggregating





groups of cohorts into representative 'patches' (conceptually similar to a forest plot) that explicitly tracks the time between disturbances (*Moorcroft et al.*, 2001). The main property of the ED concept that differs from most commonly used 'big-leaf' models is the capacity to predict distribution, structure, and composition of vegetation directly from their given physiological traits described by the model parameterization (*Fisher et al.*, 2015). This is achieved via the means of trait-filtering, whereby

plant traits affect plant growth and survival, growth in turn affects the acquisition of light resources, and feeds back onto growth, survival and reproduction. Differences in growth, survival and reproduction rates thus directly control the relative distributions of vegetation types and their traits as well as the overall carbon stocks. CLM4.5(ED) can be simulated with different modes including point mode for individual sites, regional mode for watershed or regional scales, and global mode for the global scale. See supplementary model description in *Fisher et al.* (2015) for details on specific components of the

model structure.CLM4.5(ED) represents vegetation using size-structured groups of plants (cohorts) which co-exist on various successional trajectory-based land units. CLM4.5(ED) simulates growth by integrating photosynthesis across different leaf layers for each cohort, and stress-based mortality by plant carbon starvation and hydraulic failure (*Fisher et al.*, 2015), in addition to a background mortality rate, tree-fall impact mortality, and fire. The model allocates photosynthetic carbon to different tissues such as leaf, root and stem based on the allometry of different tree species.

In this original version of CLM4.5(ED), there are two challenges for the model to simulate tropical forests. First, it is difficult for the model to represent the coexistence of PFTs due to the lack of representation of plant trait trade-off between different PFTs. Therefore, we focus only on a single broadleaf evergreen tree PFT, which is a typical vegetation type for the study region (the Amazon). We want to point out that, because of the high species diversity of tropics, it is always a challenge for models to capture diverse traits with a limited number of PFTs. Even though we have only one PFT in this study, our sensitivity analysis

will help us understand the main control on demographic rates of growth and mortality that will essentially affect the outcome of competition for multiple PFTs. Thus, we expect that our sensitivity analysis can be used to guide the selection of traits for the presentation of trait trade-off for diverse tropical forests and improve the simulation of PFT coexistence for model calibration and improvement. Second, the model generally underestimates leaf area index (LAI). We expect that our sensitivity analysis will be used as a guidance to adjust identified key model parameters in order to better fit model predictions to the observations.

The CLM4.5(ED) tracks different size class of plants (generally >10) through time. To facilitate our analysis, we aggregate cohorts into 3 size categories: small (<10 cm), medium (10-50 cm) and large trees (> 50 cm). For sensitivity analysis of each size category (small, medium and large trees), we choose to average the outputs over 30 year intervals. This is done in view that the transient and abrupt changes across different size categories in the annual model outputs could make the FAST analysis only account for a minor amount of the variance contribution from each parameter.

## 2.2  Sensitivity Analysis: The FAST method

Global sensitivity analysis aims at quantifying the contributions of input variables to the variance of the outputs of a physical model by simultaneously sampling values of parameters from their corresponding statistical distributions. There are many methods for global sensitivity analysis. Two popular variance-based approaches are the Sobol's method (*Sobol'*, 1990) and the Fourier Amplitude Sensitivity Test (FAST) (*Cukier et al.*, 1973). The Sobol's method has received much attention since it



provides clear description of the importance index of model parameters based on variance decomposition. However, the full description requires the evaluation of $2^n$ Monte Carlo integrals (*Sudret*, 2008), which is not practically feasible unless $n$ is low ($n$ here represents the dimensionality of the model, or the number of active parameters). Compared to Sobol's method, FAST is more computationally efficient. It can be used effectively for nonlinear and nonmonotonic models (*Sudret*, 2008; *Xu*

*and Gertner*, 2011). FAST uses a periodic approach to sample the parameter space and a Fourier transformation to decompose the variance of a model output into partial variances contributed by different model parameters. Only the first order sensitivity indices referring to the "main effect" of parameters were calculated in the original method. In the 1990's, an extended FAST method able to calculate sensitivity indices referring to "total effect" was developed (*Sobol'*, 1990; *Archer et al.*, 1997; *Saltelli et al.*, 1999). This "total effect" of a parameter's sensitivity refers to the sum of a parameter's individual contribution (1st order

sensitivity) and the contribution from its interaction with other parameters (higher order sensitivity) on the overall variance of the model output; that is, the total effect includes all the higher order interactions. *Xu and Gertner* (2011) further derived equations within the FAST framework to calculate specific higher order interactions for different sampling approaches. The FAST method has found widespread use in many different fields of study including sensitivity analysis of the parameters of models that represent the land surface (*Collins and Avissar*, 1994), chemical reaction (*Haaker and Verheijen*, 2004), nuclear

waste disposal (*Lu and Mohanty*, 2001), erosion (*Wang et al.*, 2001), hydrologic systems (*Francos et al.*, 2003), atmospheric systems (*Kioutsioukis et al.*, 2004), crop growth (*Wang et al.*, 2013), or matrix population and forest landscape models (*Xu and Gertner*, 2009; *Xu et al.*, 2009).

In this study, we quantify both the 1st and 2nd order sensitivities of the model parameters using FAST. It is possible to identify higher order interactions with FAST; however, because of the sample size limitations for a larger tri-variate parameter space,

the FAST-based estimation of third-order sensitivity indices would be less reliable (*Xu and Gertner*, 2011). Specifically, the 1st order sensitivity is used to measure the importance of the variations in one parameter to the model outputs. If one parameter $x_i$ is important to a model output $y$ at time $t$ [i.e., $y(t)$], we expect that the mean value of $y(t)$ will change substantially with different values of $x_i$. Statistically, we expect to see a large variance of the expected value of $y(t)$ given $x_i$ [i.e., large $V\big(E\big(y(t)|x_i\big)\big)$, where $E(\cdot)$ is the expected value of the output, $V(\cdot)$ is the variance calculated in the parameter space].

Similarly, if the combined impact of $x_i$ and $x_j$ is important, we expect to see a large variance of the expected value of $y(t)$ given $x_i$ and $x_j$ [i.e., large $V\big(E\big(y(t)|x_i,x_j\big)\big)$]. Therefore, we calculate the 1st and 2nd order sensitivities, $\alpha_{x_i}$ and $\alpha_{x_ix_j}$, respectively, of the model parameters for each output of interest and at each time step as follows,

$$\alpha_{x_i}(t) = \frac{V\big(E\big(y(t)|x_i\big)\big)}{V(y(t))} \tag{1}$$

$$\alpha_{x_i,x_j}(t) = \frac{V\big(E\big(y(t)|x_i,x_j\big)\big) - V\big(E\big(y(t)|x_i\big)\big) - V\big(E\big(y(t)|x_j\big)\big)}{V\big(y(t)\big)}, i \neq j. \tag{2}$$

where $V\big(y(t)\big)$ is the total variance of model output $y(t)$. In FAST, the variances are estimated through periodic samples in the $\theta$ space between 0 and $2\pi$, which are linked to the samples in the parameter space through a search function. Further details on





the FAST toolbox used for this study can be found in *Xu and Gertner* (2007, 2009, 2011) or *Xu et al.* (2009). We are aware that FAST can provide robust estimates of the sensitivity coefficients in high-dimensions (*Wang et al.*, 2013), especially since the CPU-demands of CLM4.5(ED) mandates application of a method like FAST due to its ability to derive sensitivity values with sparse sampling. This allows FAST to compute with less forward runs from the original model. What is more, the parameter

space used in this study is confined to a relatively small area around the default CLM4.5(ED) values (see next section). This further assures that the sampling is sufficient to capture accurate sensitivity values.

To better understand how parameters affect specific CLM4.5(ED) output variables, we also fitted cubic splines to the scatter-plots between samples of parameters identified as important by FAST and the corresponding output variable of interest using the R SemiPar package (*Ruppert et al.*, 2003).

**2.3    Parameter Selection**

There are more than 200 parameters in CLM4.5(ED). In this study, we focus solely on vegetation components with 87 parameters that are relevant to vegetation processes, including parameters for photosynthetic processes, temperature response, allometry description, radiative transfer, recruitment, turnover and mortality. See Table D1-D4 in the appendix for a complete list of the parameters used in this study, with corresponding description, units, default values, and applied ranges. Refer

to Appendix A for the allometry equations, Appendix B for the temperature response curve (photosynthesis) equations, and Appendix C for the carbon storage equations used in CLM4.5(ED).

We use a broad definition of parameter and extract numerous features of the model that were 'hard-wired' in previous CLM versions, such as the parameters defined in the equations of Appendices A and B. The FAST algorithm requires valid ranges to be chosen for each parameter, which creates the possible parameter space to sample from. In theory, each parameter has a

corresponding observational distribution that produces the ideal space for sampling (*LeBauer et al.*, 2013). However, in this study there are both a large parameter set and a scarcity of appropriate data sources for Amazonian forests for many of the relevant quantities, therefore obtaining a robust data-supported distribution for each parameter was difficult. Because we only aim to understand the baseline model structure, the parameter ranges in this study were generated by applying a uniform distribution over a range that spans +/- 15 % of the default parameter values of CLM4.5(ED) (i.e. default parameter values

for tropical evergreen trees). We choose a rather conservative range of +/- 15% of the default CLM4.5(ED) values so that global sensitivity indices can be estimated in the reasonable vicinity of the default parameters. We suggest that a more robust uncertainty analysis based on realistic parameter ranges is needed for guidance on additional field measurements.

Using FAST, 5000 parameter combinations are sampled from the parameter space. The sample size was determined using the method of *Xu and Gertner* (2011) where it is appropriate to use 100 times the number of effective (important) parameters.

The 5000 model runs cost about 32 CPU hours for each simulation, and thus we ran our simulations for a total of 160,000 CPU hours on the Los Alamos National Laboratory (LANL) Conejo super computer.

In this analysis, we assume the majority of CLM4.5(ED) parameters to be non-correlated with uniform probability due to the limitation of data for covariate traits for the 80+ parameters in this study. However, we do need to take care of the correlation among parameters in the temperature response functions (Appendix B) in order to generate realistic temperature





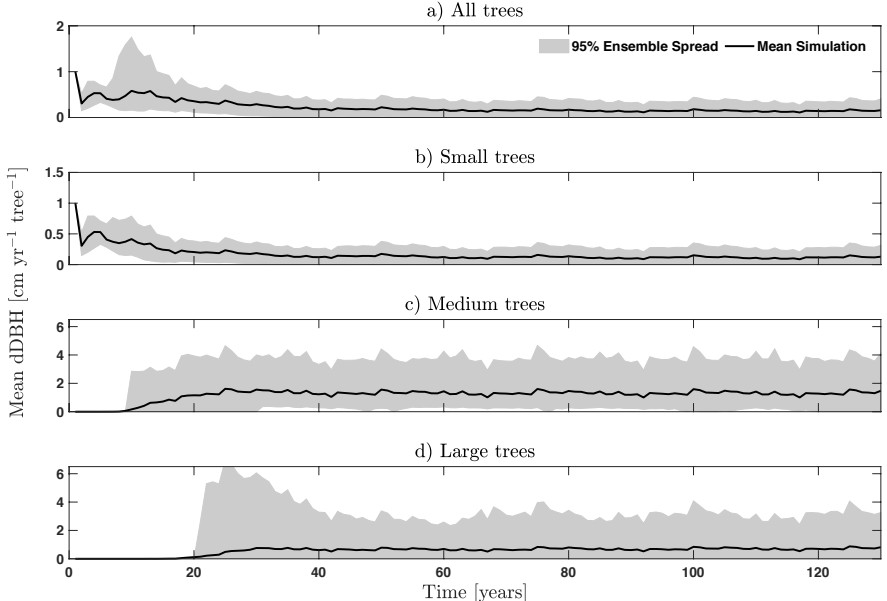

**Figure 1.** Simulated change in diameter at breast height (dDBH; cm yr$^{-1}$ tree$^{-1}$) for a) all trees, b) small ($diameter < 10$ cm), c) medium (10 cm $< diameter < 50$ cm), and d) large trees ($diameter > 50$ cm). Shown are the mean simulation (black line) with 95% spread of the simulation ensemble.

response curves. These parameters are tested for correlation using a published dataset (*Leuning*, 2002), which showed that the photosynthetic parameters for activation energy (e.g. $V_{c,max,ha}$) are not necessarily correlated with the other photosynthetic parameters. However the parameters for deactivation energy (e.g. $V_{c,max,hd}$) and those related to entropy terms (e.g. $V_{c,max,se}$) are highly correlated as expected (correlation = 0.99+). Thus, each of these parameters' samples are generated from the same

5 location in their relative parameter spaces, which maintains their correlation.

## 2.4 Data and Model Setup

In this study, the CLM4.5(ED) model simulations are setup for a $1°$ by $1°$ grid in a moist-tropical forest in the State of Pará, The Amazon, Brazil ($7°$ S, $55°$ W), which is a default tropical setup for CLM. We initialized the runs with a bare ground, or a state with no vegetation, and simulated the forest dynamics for 130 years, which we determined is enough time for the ecosystem

10 to reach equilibrium dynamics since the simulated outputs and corresponding sensitivity values for biomass, basal area and various carbon fluxes had stabilized by this time. Therefore, the specific behavior of our demographic model emerges in a shorter time scale, i.e. 130 years, compared to what would naturally occur for a forest to reach stable states from bare ground in reality. By choosing to start from bare ground and running the model until it reaches a quasi-steady-state size distribution, rather than by examining short runs initialized from observed initial forest size distributions (e.g. *Dietze et al.* (2014)), we





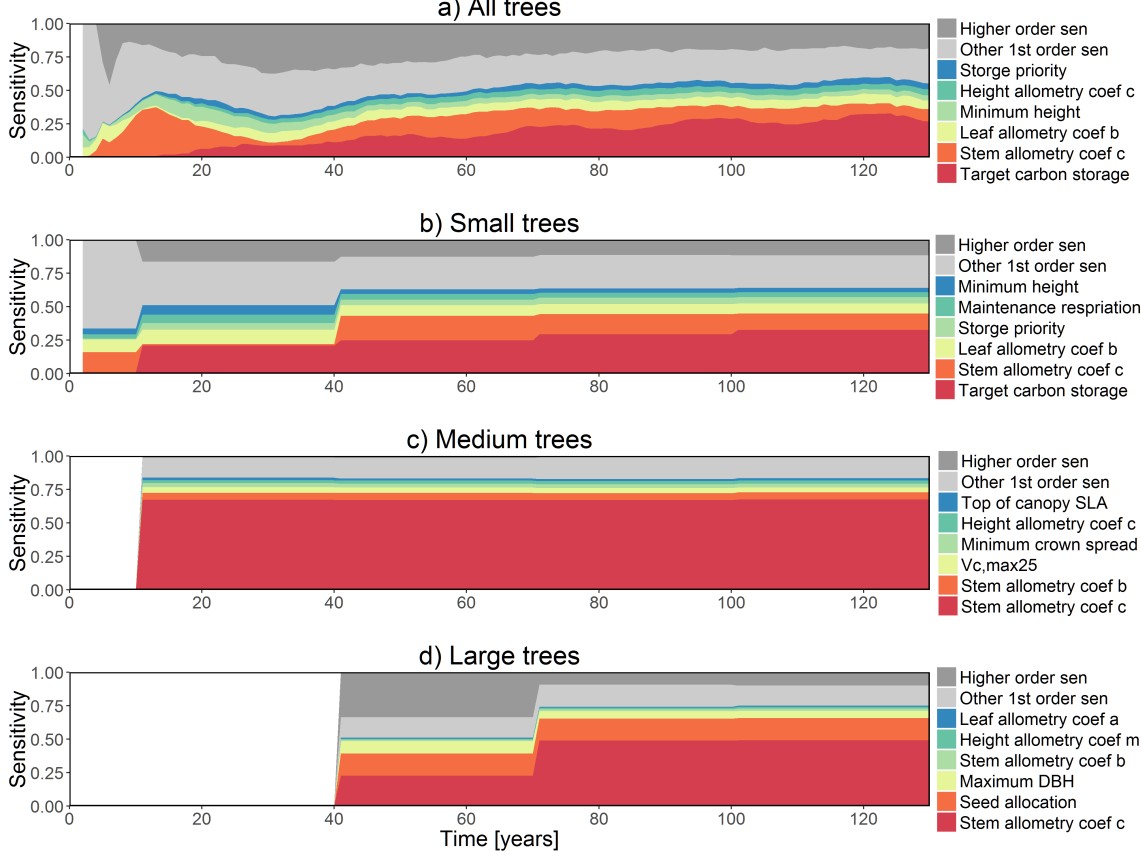

**Figure 2.** Sensitivity index of the model parameters (1$^{st}$ order) for change in diameter at breast height (dDBH) outputs from CLM4.5(ED) (outputs of dDBH are in units [cm yr$^{-1}$ tree$^{-1}$] and sensitivity index is unitless) for a)all trees, b)small trees ($diameter < 10$ cm), c) medium trees (10 cm $< diameter < 50$ cm), and d) large trees ($diameter > 50$ cm). Shown are the top 6 most important parameters, in order of importance (red is the most important and blue is the least important). The jumps seen in years 10, 40, 70, and 100 for small, medium and large trees are due to the temporal averaging mentioned in the materials and methods section. The figures also show sensitivities of the remaining parameters in light grey (1st order sensitivity index for all other parameters) as well as the sensitivity of parameter interactions in dark grey (higher order sensitivity index for all parameters).

are deliberately allowing the ecosystem demographic structure itself to be an outcome of the parametric variance rather than a separate, possibly non-self-consistent, initial condition variance. The climate conditions for this site are from *Qian et al.* (2006) representative of data from 1948-1972 and recycled for the 130 year simulations.





# 3   Results

In this section, we highlight the outputs of CLM4.5(ED) from the 5000 simulations obtained for the FAST analysis, and then show the important parameters that control variance in the outputs. We first investigate the forest demographic dynamics, diagnosing the growth and mortality processes simulated in CLM4.5(ED), i.e. outputs representing the change in diameter at breast height (dDBH), the mortality rate, and the resulting basal area (BA). Then we analyze the forest carbon cycles, specifically the carbon fluxes and stocks in the model simulations, i.e. outputs representing Gross Primary Production (GPP), Net Primary Production (NPP), LAI, and total forest biomass.

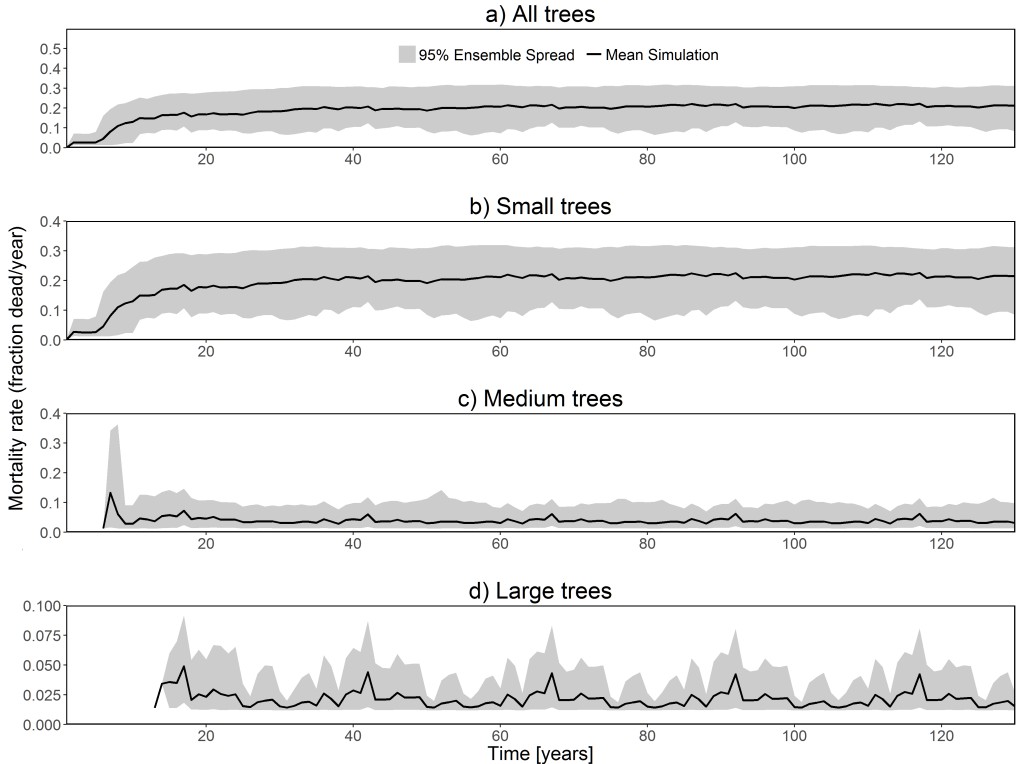

**Figure 3.** Simulated mortality rates (fraction dead per year) for a) all trees, b) small trees ($diameter < 10$ cm), c) medium trees ($10$ cm $< diameter < 50$ cm), and d) large trees ($diameter > 50$ cm). Shown are the mean simulation (black line) with 95% spread of the simulation ensemble.

## 3.1   Forest demographic dynamics: growth and mortality

One of the key properties of CLM4.5(ED) is that vegetation is represented as cohorts of varying sizes for more realistic simulation of light competition in the canopy. We group various cohorts of trees into 3 size categories for analysis purpose:



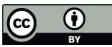

small, medium , and large trees. Since the model runs are initialized from a near-bare ground state, all simulated plants are considered 'small' with an initial density of half-centimeter diameter saplings.

For the stem growth in terms of dDBH averaged per tree (Fig. 1), small trees have lower rates of growth compared to medium or large trees as they are mostly in the understory with a lack of light for growth. However, the fraction of overall stem diameter

growth is dominated by the small trees (Fig. D1) due to their high densities (Fig. D2). Our first-order sensitivity analysis based on FAST shows that the most sensitive parameters for tree growth are the target storage carbon and stem allometry parameters, however, the importance magnitude varies for different sizes of trees (Fig. 2). We observe that the stem allometry coefficient c is the dominant parameter that controls dDBH for medium and large trees, and the target carbon storage is the most important parameter for small trees. A higher value of stem allometry coefficient c, or a higher allocation of carbon to stem, will lead to

a larger allocation of carbon to stem and thus a faster DBH growth in the initial life stage of small trees (Fig. D3). However, for medium and large trees, a higher allocation of carbon to stem can lead to lower proportion of carbon allocated to leaves for productivity and thus a slower DBH growth (Fig. D3). This support hypothesis H2. The target carbon storage determines the target amount of carbon for the plant to store relative to the leaf biomass (see Appendix C for details). Smaller trees have less stem biomass and are less impacted by the stem allometry coefficient c parameter. Furthermore, small trees are vulnerable

to changes in the amount of target carbon storage which affects carbon allocation to growth (see Eq. A9 in Appendix C). Our sensitivity analysis also shows specifically important parameters for different sizes of trees. For example, leaf allometry is important for small trees, $V_{c,max25}$ for medium trees, and seed allocation for large trees.

Mortality is an important driver for the simulated forest dynamics in CLM4.5(ED). It includes four modes of mortality: 1) fixed background mortality, 2) hydraulic failure based on a threshold of very low soil moisture; 3) carbon starvation resulting

from the depletion of carbon storage in plants (see Appendix C for details); and 4) impact mortality resulting from the falling of big trees (*Fisher et al.*, 2015). In the model, carbon starvation is the main driver for the overall mortality (Fig. D4 in the appendix). Carbon-starvation-based mortality uses a threshold of carbon storage to trigger mortality (see Appendix C). Under shaded conditions, less carbon storage as determined by target carbon storage, respiration, and NPP could lead to a higher mortality rate. As expected, the smaller tree size classes represent a much larger fraction of the total mortality (Fig. 3). The first-

order sensitivity analysis of predicted mortality rate (percentage of mortality per year) shows that the dominant parameter for predicting mortality of large trees is the target carbon storage (Fig. 4); however, for small and medium trees, other parameters such as allometric and photosynthetic parameters that could potentially determine their competitive advantages in the canopy are also important (Fig. 4). Specifically, for medium size trees, the mortality rate is affected by both the stem allometry coef c and targeted carbon storage (Fig. 4). For the small trees, important parameters include the photosynthetic capacity parameter

($V_{c,max25}$), stem allometry coefficient c, mortality rate under stress, and maintenance respiration, with the target carbon storage having high sensitivity for small trees in the early years.

The simulated BA of the forest, which is the total stem cross-sectional area per ground surface area, results from the combination of both DBH growth and mortality. The BA reaches equilibrium for different sizes of trees around year 70 (Fig. 5). Our FAST analysis shows that a key parameter that controls BA in different sizes of trees is the stem allometry coefficient c

(Fig. 6), which is a major parameter that determines the DBH growth (Fig 2). Meanwhile, the target carbon storage parameter





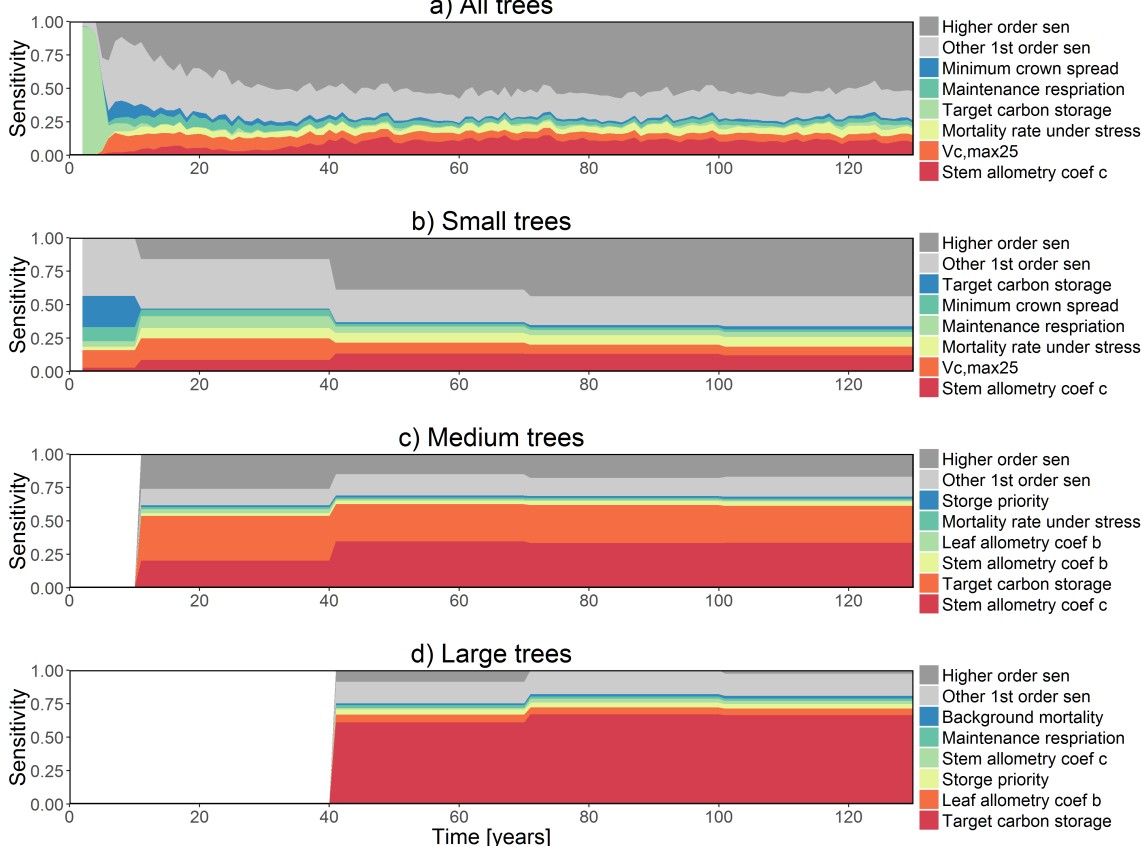

**Figure 4.** Sensitivity index of the model parameters (1$^{st}$ order) for the mortality outputs of CLM4.5(ED) (outputs for mortality are in units [$fraction$ yr$^{-1}$] and sensitivity index is unitless) for a) all trees, b) small trees, c) medium trees, and d) large trees. Shown are the top 6 most important parameters, in order of importance (red is the most important and blue is the least important). The jumps seen in years 10, 40, 70, and 100 for small, medium and large trees are due to the temporal averaging mentioned in the materials and methods section. The figures also show sensitivities of the remaining parameters in light grey (1st order sensitivity index for all other parameters) as well as the sensitivity of parameter interactions in dark grey (higher order sensitivity index for all parameters).

that dominantly controls mortality is also an important parameter for the simulated BA (Fig 6). A new parameter that becomes important for BA of small and medium trees is the minimum crown spread, which determines the ratio of crown radius to DBH. A larger crown spread can lead to a smaller number of trees in the canopy and thus a lower BA (Fig. D5).

For the second order sensitivity analysis, parametric interactions between stem allometry coefficient c and the proportion of carbon for seed allocation and target carbon storage, are found to be important for the prediction of total BA (Fig. D6). For trees of different sizes, parametric interactions between stem allometry coefficient c and three other parameters, which are the minimum crown spread, target carbon storage, and maximum DBH parameters, are important for small, medium and large trees, respectively. For the prediction of dDBH and mortality, the contribution of most parametric interactions are relatively





small except for large trees (Fig. D7). The interactions between stem allometry coefficient c and the proportion of carbon for seed allocation, maximum DBH, and stem allometry coefficient b parameters, are important for the prediction of dDBH for large trees. For the prediction of mortality, the interaction between stem allometry coefficient c and target carbon storage is found to be important for large trees (Fig. D8).

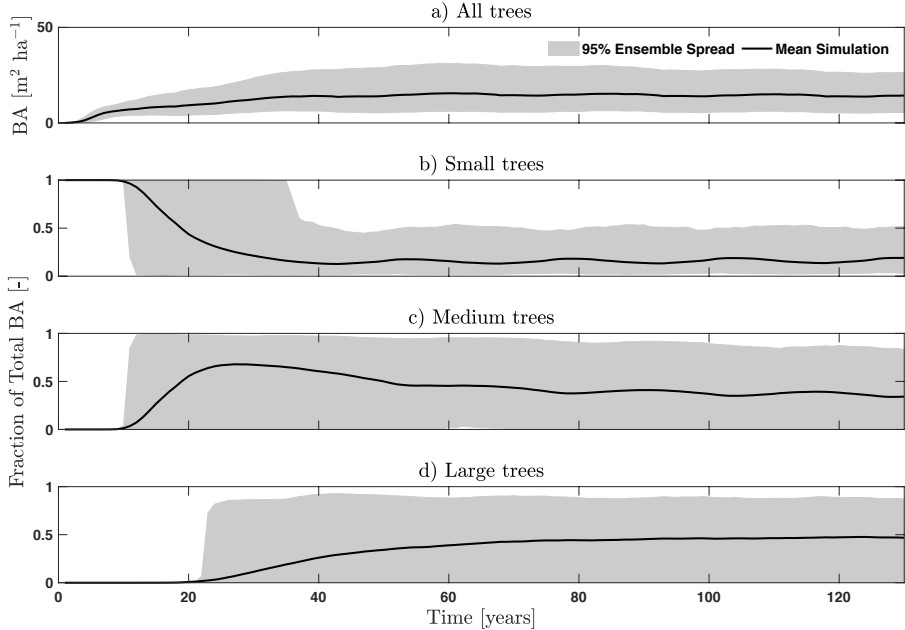

**Figure 5.** Simulated basal area (BA, units are in m$^2$ ha$^{-1}$) from CLM4.5(ED) for a) all trees, and its fractional distribution for various tree sizes classes, including b) small ($diameter < 10$ cm), c) medium (10 cm $< diameter < 50$ cm), and d) large trees ($diameter > 50$ cm). Shown are the mean simulation (black line) with 95% spread of the simulation ensemble. The fractional outputs shown in Panels B-D are the percentage of total BA that is associated with each tree size.

## 3.2 Forest carbon cycles: carbon fluxes and stocks

To investigate the key parametric control on carbon fluxes and stocks, we specifically investigate parameter sensitivities for GPP, NPP, LAI, and total forest biomass. Our results show that GPP and NPP increased consistently for the first 10 years of the simulations, which is expected for a forest growing from bare ground (Fig. 7). However within a fairly short period of 5-10 years, GPP, NPP and LAI and their variance reached a quasi-stable rate. This amount of time to reach equilibrium is much shorter compared to the basal area (Fig. 5a) and the total biomass accumulations (Fig. 7d).

The first-order sensitivity analysis based on FAST shows that, for carbon fluxes of GPP and NPP, the photosynthetic capacity parameter ($V_{c,max25}$) is the most sensitive parameter (Fig. 8), which supports our hypothesis H1. This result is not surprising as this parameter controls leaf-level carbon uptake. Interested readers should refer to *Fisher et al.* (2015) for details on how



$V_{c,max25}$ is involved in calculation of production and respiration of simulated forests in CLM4.5(ED). Furthermore, specifically for NPP, the respiration parameters such as the growth and leaf maintenance respiration show high sensitivity (Fig. 8). For LAI, the leaf allometry parameter is the most important as it determines carbon allocation for leaves. The stem allometry parameter is the most important for total biomass as it determines carbon allocation to the stem, which supports our hypothesis H2. A

common sensitive parameter is the target carbon storage which is important for GPP, NPP, LAI and total biomass. This results from the fact that the target carbon storage is a key driver for mortality especially for medium and large trees in the simulations (Fig. 4), which account for a large proportion of biomass (Fig. D9 in the appendix) and GPP (Fig. D10). This result supports hypothesis H3. For the second order sensitivity, the contribution of most parametric interactions are relatively small (Fig. D11) as the first-order sensitivity accounts for a majority of the total variance in model outputs (Fig. 8).

Our bi-variate spline analysis (*Wahba*, 1990) shows that, for $V_{c,max25}$ and target storage carbon, an increase in either of these parameters will cause an increase in the output of GPP, NPP, LAI and biomass (Fig. 9). For the parameters related to leaf and stem allometry, however, the relations may differ depending on the output and the year of interest. At year 130, the higher leaf allocation normally leads to higher fluxes (NPP and GPP) but less biomass. Meanwhile, higher stem allocation lead to higher biomass but smaller fluxes (NPP and GPP). This suggests that the trade-offs between carbon allocation to stem vs leaf tissues

leads to a corresponding tradeoff between carbon stocks and productivity in the model predictions.

## 4   Discussion

### 4.1   Comparing parameter sensitivities to other models

While second generation vegetation demographic models such as CLM4.5(ED) provide us great opportunities to predict carbon cycles with LSMs, the larger number of parameters also creates challenges for identifying key processes for further investiga-

tion. In comparison with previous sensitivity analyses of size-structured models, our study considers a much larger number of parameters, i.e. $> 80$ compared with $\sim$20-35 parameters (*Pappas et al.*, 2013; *LeBauer et al.*, 2013; *Wang et al.*, 2013; *Dietze et al.*, 2014). In general, our analysis shows similar results to sensitivity analysis on first generation 'big-leaf' vegetation models (e.g., *Sargsyan et al.* (2014)) in view that photosynthetic capacity, $V_{c,max25}$, is a key parameter for predicting GPP and NPP fluxes. However, we do show important parameters that are unique to LSMs with second generation vegetation demography.

Specifically, results shown here indicate the importance of leaf and stem allometry parameters, which control dynamic carbon allocation strategies based on size, and thus control the general vegetative state and size structure of the forest (*Waring et al.* (1998); *Waring and Running* (2010)). The importance of allometric parameters could also result from the fact that the relationship between allometric coefficient c and carbon allocation is highly non-linear based on a power function (see Appendix A for details). Our sensitivity analysis also shows the importance of carbon storage to the prediction of mortality rate and thus

the total biomass. This is in agreement with other sensitivity analyses of CLM which show the plant mortality rate as a key parameter for the prediction of total biomass (*Sargsyan et al.* (2014)).

Specifically, we found consistencies in which parameters are sensitive for simulated carbon fluxes and stocks, which are the photosynthetic capacity, mortality and respiration parameters (*Pappas et al.*, 2013; *LeBauer et al.*, 2013; *Wang et al.*,





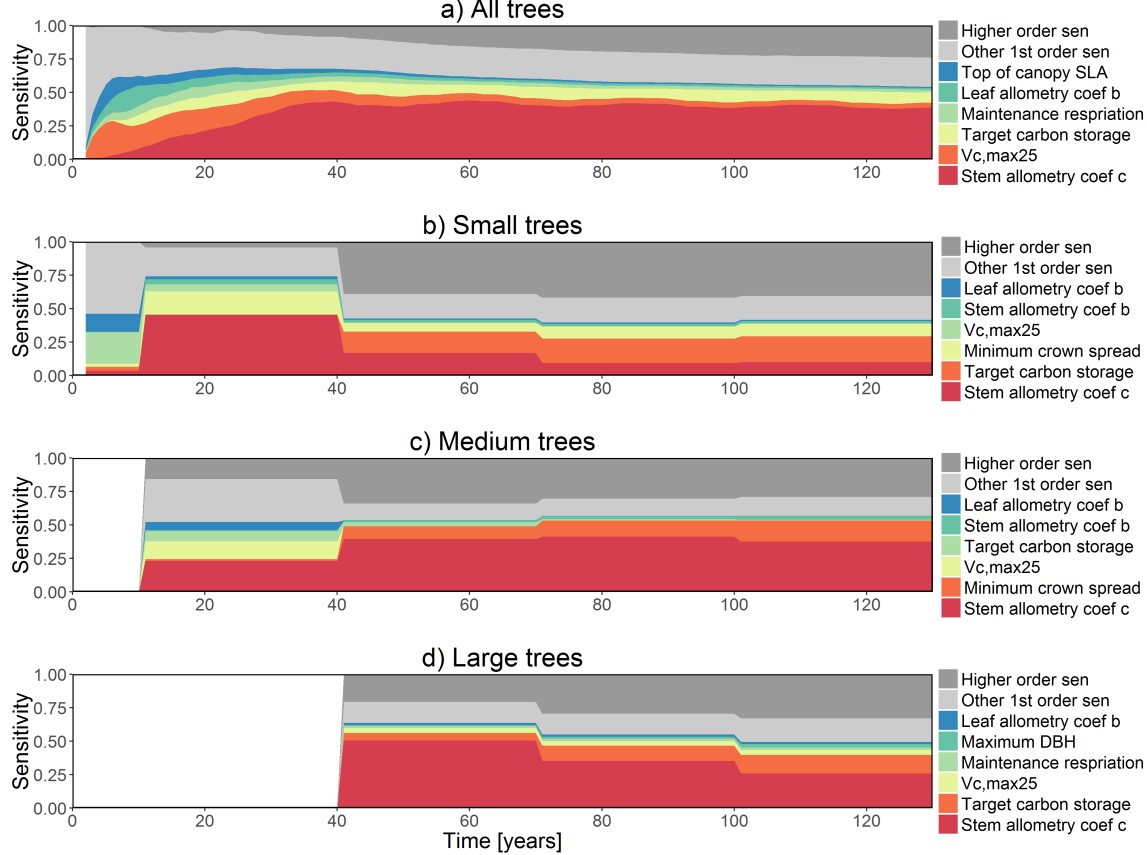

**Figure 6.** Sensitivity index of the model parameters (1$^{st}$ order) for basal area (BA) outputs from CLM4.5(ED) (outputs of BA are in units [m$^2$ ha$^{-1}$] and sensitivity index is unitless) for a) all trees, b) small trees ($diameter < 10$ cm), c) medium trees (10 cm $< diameter < 50$ cm), and d) large trees ($diameter > 50$ cm). Shown are the top 8 most important parameters, in order of importance (red is the most important and blue is the least important). The jumps seen in years 10, 40, 70, and 100 for small, medium and large trees are due to the temporal averaging mentioned in the materials and methods section. The figures also show sensitivities of the remaining parameters in light grey (1st order sensitivity index for all other parameters) as well as the sensitivity of parameter interactions in dark grey (higher order sensitivity index for all parameters).

2013; *Dietze et al.*, 2014); however, there are differences in the order of parameter importance. For example, *Dietze et al.* (2014) showed that growth respiration fraction is the most important parameter for the simulation of NPP, and $V_{c,max25}$ only ranked as the 7th most important parameter. For our analysis, $V_{c,max25}$ and growth respiration fraction are the first and second most important parameters. This difference in parameter sensitivity rank may result from the fact that *Dietze et al.* (2014) used variable parameter ranges based on data while our sensitivity analysis uses equal percentage variations (see details in the discussion subsection: *Limitation of methods*). We also found that some parameters that are identified as important in other studies are not found to be important in our analysis. For example, *Dietze et al.* (2014) showed that water conductance





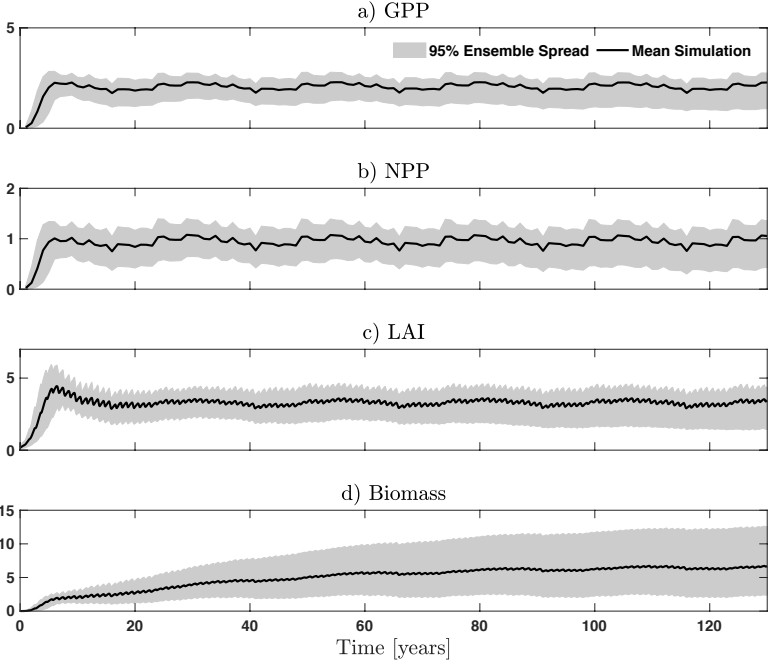

**Figure 7.** Simulated GPP and NPP (in kg C m$^{-2}$ year$^{-1}$), LAI (in m$^2$ m$^{-2}$), and biomass (in kg C m$^{-2}$) from CLM4.5(ED) model. Shown are the mean simulation (black line) with 95% spread of the simulation ensemble. The system is initialized with a bare ground, and this is shown with initial values of 0 for the different outputs.

that determines the upper boundary of transpiration is the second most important parameter for simulated NPP, but a similar parameter (smpso; Table D2) that defines soil water potential for opening stomata is not important in our analysis. This could be related to the fact that our site is much wetter than the temperate forests simulated by *Dietze et al.* (2014). *Pappas et al.* (2013) showed that the root distribution parameter that determines the fraction of fine roots in the upper soil layer is one of the top five parameters for the simulations of vegetation carbon fluxes and stocks; however, in our sensitivity analysis, the two root distribution parameters ($root_a$ and $root_b$: Table D2) are not important for both vegetation carbon fluxes and stocks. This difference could also result from a wider range of variations (∼+/-30%) in the study of *Pappas et al.* (2013) compared to our 15% variations of the default parameters. Finally, our analysis shows the importance of allometry parameters, which are not considered in previous studies (*Pappas et al.*, 2013; *LeBauer et al.*, 2013; *Wang et al.*, 2013; *Dietze et al.*, 2014). Due to the large uncertainties that are associated with allometry (*Dietze et al.*, 2008), it would be important to consider better parameterization of allometry for second-generation vegetation demographic models within LSMs.

## 4.2 Comparing simulations with observations

The goal of our study is not to reproduce the observations but instead to identify important parameters that can be better estimated for the model to fit observations. We do want to highlight two caveats. First, the improved estimation of the most





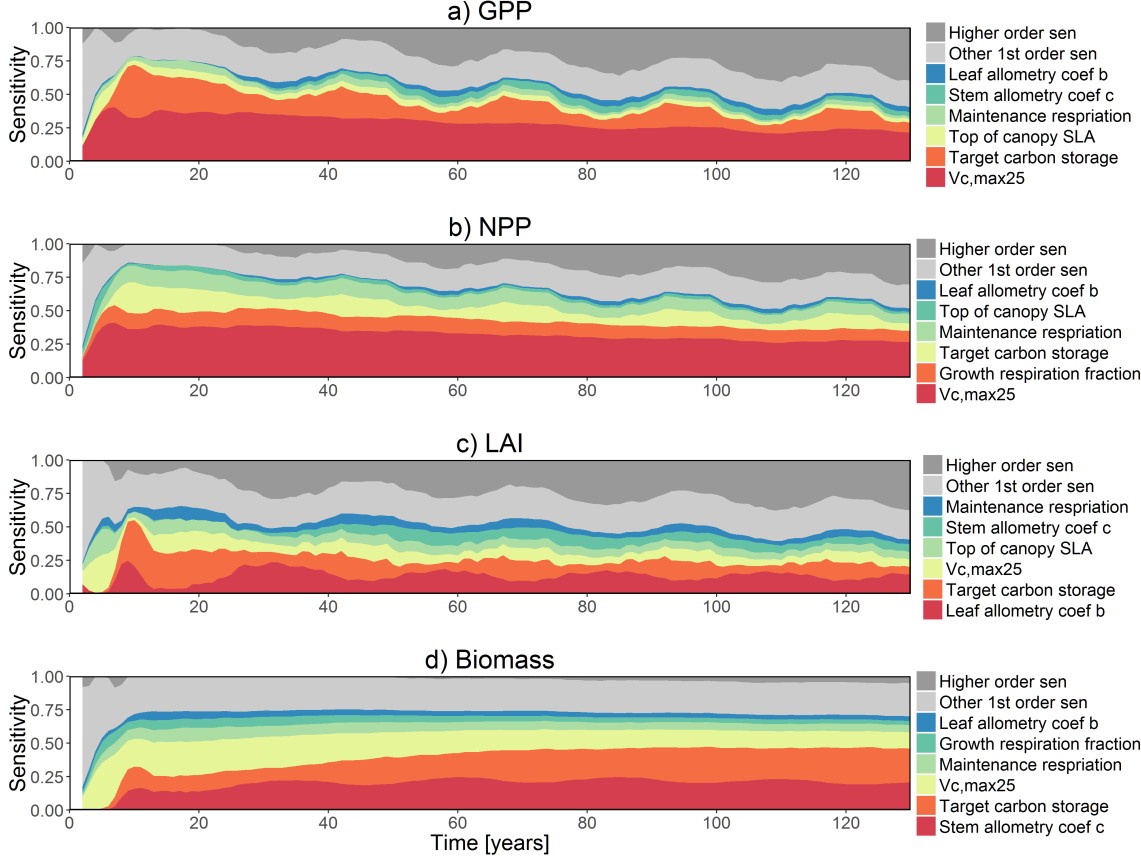

**Figure 8.** Sensitivity index of the model parameters (1st order) for a) GPP, b) NPP, c) LAI, and d) biomass (units for each output are shown in Fig. 7 and sensitivity index is unitless). Shown are the top 6 most important parameters, in order of importance (red is the most important and blue is the least important). The figures also show sensitivities of the remaining parameters in light grey (1st order sensitivity index for all other parameters) as well as the sensitivity of parameter interactions in dark grey (higher order sensitivity index for all parameters).

sensitive parameters may not be most efficient if they have relatively small uncertainty or variability across different species and locations. Second, even if the estimates for most sensitive parameters are perfect, we may still not be able to fit model predictions to observations if there is deficiency in the representation of key processes in the model. Given observation data limitations for our site, we conduct a qualitative comparison of our model simulations to ranges reported in the literature for

5     the tropics. Not surprisingly, our model results show a variation of model-data mismatch for key vegetation states. For LAI (Fig. 7c), our simulated range is between $\sim$1.9-6.0 m$^2$ m$^{-2}$ which is lower than the observed range of $\sim$3.0-6.9 m$^2$ m$^{-2}$ based on LAI estimated from MODIS imageries (*Knyazikhin et al.*, 1999) during 2000-2016 within a 0.5 degree window around our site. Our sensitivity analysis showed that leaf allometry coef b and target carbon storage are two key parameters affect simulated LAI (Fig. 8) and we expect that a better estimation of these parameters with data could potentially improve the

10    model simulations. For GPP (Fig. 7a), the simulated range is between $\sim$1.0-3.0 kg C m$^{-2}$ yr$^{-1}$, which is also lower than the





observed range of ∼2.4-3.7 kg C m$^{-2}$ yr$^{-1}$ based on extrapolation from eddy fluxes tower observations and climate during 1981-2010 (*Jung et al.*, 2009). Our analysis suggests that photosynthetic capacity as represented by $V_{c,max25}$ , target carbon storage and top of canopy specific leaf area are important parameters (Fig. 8) and an improved estimation of them could help improve model simulations of GPP. We are not able to access onsite data for other model outputs. Therefore, we compare our

model outputs with ranges from multiple tropical sites to evaluate their validity. For biomass (Fig. 7d), the simulated range of ∼2.5-12.5 kg C m$^{-2}$ is lower than the observed range of ∼7.3-21.3 kg C m$^{-2}$ from 21 transects within 3 tropical sites (*Hunter et al.*, 2013). For BA (Fig. 1a), the simulated range of ∼5.0-30.0 m$^2$ ha$^{-1}$ is also lower than the observed range of ∼17.1-35.2 m$^2$ ha$^{-1}$ from 5 tropical sites (*Hunter et al.*, 2013). Our results show that stem allometry coef c is the most important control on BA and biomass, and an improved parameterization on stem allometry could help improve the model simulations. For the

DBH growth, there are large variances in the observed values across different sites with the range of 0-3 cm/year (*Lieberman et al.*, 1985; *Worbes*, 1999; *Adams et al.*, 2014). The simulated average DBH growth is between 0-0.4 cm/year but could be as high as 4 cm/year for medium and large trees (Fig. 3). Based on our sensitivity analysis (Fig. 4), we expected an improved parameterization of both allometry coef c and target carbon storage could help fit the model predictions to data.

We compare our mortality simulations with an extensive dataset of observed mortality of 1781 species from 14 pan-tropical

large area ForestGEO forest dynamics plots (*Johnson et al.*, 2018). For this study, the forest plots ranged from 2 to 52 ha each with 371 ha in total in which all recorded stems are ≥ 1 cm diameter at breast height. Our comparison shows that the CLM4.5(ED) simulations of medium and large tree mortality (Fig. 5CD) are close to the 95% confidence interval of observed values, which is about ∼0.5-5.7% per year. However for small and medium trees (Fig. 5BC), the simulated mortality rate of ∼15-30% and ∼1-10% is high when compared to the observed 95% confidence interval of mortality rate of ∼0.6-11.3% and

∼0.8-3.0% for small and medium trees, respectively. The high predicted mortality rate of small trees could result from the fact that the model predicts a very high mortality rate for very small trees (<1 cm) as they cannot establish themselves due to low light conditions in the simulations. Since the small trees have such a large fraction of the population in our simulations (see Fig. D2), the overall mortality rate (Fig. 3A) of ∼15-30% is also high when compared to observations (∼0.6-11.3%); however, if we separate the mortality rate of very small trees from the calculation of the overall mortality, then the simulated mortality

rates of  1-10% (Fig. D12) are in the range of observations. The very high mortality rate range of smaller trees (∼10-30%; Fig. D12) spans the reported seedling/sapling mortality rate, e.g., ∼15-21% per year from 1-20 year old tropical forest stands in Costa Rica (*Dupuy and Chazdon*, 2006). However, there is potential for improvement for site-level simulations as the current recruitment algorithm within CLM(ED) depends only on the availability of seed bank but not on the density, light and water availability. The relatively high mortality rate of small and medium trees could also be linked to the fact that CLM(ED) uses the

perfect plasticity approximation (PPA) to simulate the canopy light availability for understory trees (*Fisher et al.*, 2018), which may create canopy closure too fast for the small and medium size trees to survive under low light conditions. We expect that future improvements on recruitment and PPA could be helpful for a better prediction of tree mortality for small and medium size trees. Our sensitivity analysis indicates that key model parameters that can be better estimated for improved mortality predictions include stem allometrey parameters, $V_{c,max25}$, target carbon storage, and mortality rate under stress (Fig. 4).



Another reason for the data-model discrepancy could result from the limited representation of diverse tropical species or traits with the simulation of one single PFT. This is a limitation of many LSMs as they typically only have 2-3 PFTs for tropical forests (e.g. only evergreen and deciduous for tropical trees within CLM). CLM(ED) has the potential to better represent the trait diversity through trait filtering under different environmental conditions (*Fisher et al.*, 2015). One critical component to

incorporate traits into the model is to represent the trait trade-off and coordination for different PFTs. Through our sensitivity analysis, we have identified key parameters for vegetation dynamics, which can be targeted for the representation of trait trade-off and coordination in the tropics. For example, our study shows that a higher stem carbon allocation could reduce the GPP and while a higher $V_{c,max25}$ could increase GPP (Fig. 9). The potential exploration of trade-off and coordination between these two parameters could be critical to resolve different PFTs to represent the trait variations. Even though the simulated ranges

of the model outputs are different than the observations, our sensitivity analysis should still be valid in view that a primary end-goal of this research is to identify important parameters that can be better estimated for the model to better fit observations. For example, Holm et al (2018, In Review) utilize results from our study to implement their tropical forest parameterization, specifically by increasing their target carbon storage parameter to obtain higher survival and thus lower growth.

In addition to directly comparing the model outputs to observations, we want to highlight that the sensitivity analysis will

also allow us to explore the functional relationships between model parameters and outputs. Future synthesis studies that show these functional relationships using data across different sites could be very useful to evaluate the fidelity of model structure to represent the key processes that control these relationships.

## 4.3 Limitation of methods

Our study is the first global sensitivity analysis for CLM4.5(ED); however, it is subjected to several limitations that could be

improved for future studies. First, our study uses an arbitrary choice of parameter ranges (+/- 15%), which determines the variance in the model outputs and the corresponding results of the sensitivity analysis. However, we expect that our analysis can reveal the importance of parameters given equal percentage of variations, which can help us gain a better understanding of the model structure. We do acknowledge that uncertainty analysis studies that specifically consider the potential variable ranges of values in the tropical forests based on observations could provide insights on what additional measurements are needed to

explain variance in the model prediction.

Second, we only consider the correlation in pairs of parameters that determine temperature responses for deactivation energy and entropy in photosynthesis. We do want to point out that the potential correlation among other parameters, such as the trade-off between mortality and growth parameters and the correlation among coefficients for allometric equations, could affect the simulated model output ranges and the sensitivity results. However, our exploration of parameter sensitivity assuming their

independence could still help us understand the baseline parameter control on model behaviors (*Xu and Gertner*, 2009). The exploration of trade-off and coordination among different parameters requires data analysis for multiple traits of the same species. The Predictive Ecosystem Analyzer (PEcAn) framework (*LeBauer et al.*, 2013) could be a useful tool to synthesize plant trait data to estimate model parameter distributions. The challenge is that, even though there are great efforts in the research community to compile plant trait data across the globe (*Kattge et al.*, 2011a, b), there are still a limitation of data





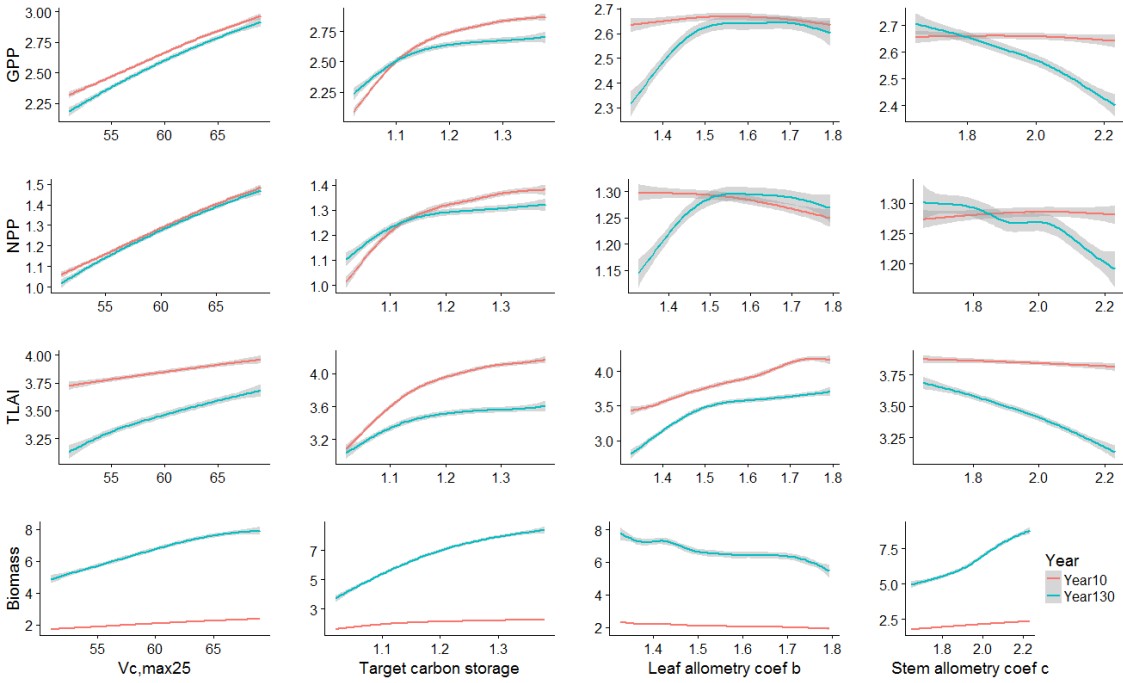

**Figure 9.** Relations between years 10 and 130 of the simulation outputs of CLM, i.e. GPP, NPP, LAI, and biomass (units shown in Figure 7), to the most sensitive parameters, i.e. $V_{c,max25}$ (unit is $umolCO_2 m^{-2}s^{-1}$), storage carbon (unit is the ratio of leaf biomass), leaf and stem allocation [unitless parameters]. Shown are the mean relations, with the 95 % spread of the mean variance in grey envelopes. These figures show how an output will generally increase or decrease when a given parameter is changed. For example, the relation between the photosynthetic parameter ($V_{c,max25}$) and all the outputs are positive, as is the case with storage carbon parameter.

with observations of multiple traits for the same species. Future uncertainty analysis studies that explicitly consider the prior distributions and correlations for all the parameters can build on this analysis and gain further insights on where the uncertainty in the model predictions come from.

Finally, it is possible that the parameter sensitivity could be different if we use different model inputs, different sites, and different structures of subcomponents within the model. For example, using site level climate drivers, instead of the reanalysis meteorological drivers used in this study (*Qian et al.*, 2006), could lead to different sensitivity values since our preliminary analysis showed that simulated vegetation demography is quite sensitive to different climate drivers. Furthermore, there are ongoing development activities to improve different components of the models. For example, there are current efforts to in-





corporate different representations of tree allometry within CLM4.5(ED), which have different formulations between size and biomass, e.g. *Chave et al.* (2014), or the current formulation of the photosynthetic process in the CLM4.5(ED) can be replaced with a model that more accurately represents the allocation of nitrogen and thus the photosynthetic process (see *Xu et al.* (2013); *Ali et al.* (2016)). Therefore, model improvements such as these can affect corresponding sensitivity analysis results.

To understand the impact of site level variations on model dynamics, similar sensitivity analysis across different sites can be conducted to understand how climate variability will affect the sensitivity analysis results.

## 5   Conclusion

LSMs have many parameters that could potentially affect the outcome of their simulations. In this study, we use the FAST to conduct a high-dimensional global sensitivity analysis on CLM4.5(ED). We use an intermediate complexity of simulation: runs

are sufficiently long to permit short-term physiological variance to propagate into the long-term forest demographic structure. Even though we do not explore competitive dynamics between different PFTs, our sensitivity analysis will guide us on the selection of key plant traits for the consideration of trait trade-off and coordination in order to improve PFT coexistence within CLM(ED).

   Our analysis show that the target carbon storage and stem allometry parameters are important for the simulation of DBH

growth for individual trees and tree mortality. The photosynthetic parameter,$V_{c,max25}$, is the most important for the simulation of carbon fluxes including GPP and NPP. The combination of stem allometry, target carbon storage and $V_{c,max25}$ dominantly control the simulation of total BA and long-term carbon stocks. The importance of growth and survival parameters in this study emphasizes the importance of understanding the dynamics of the next generation of demographically enabled LSMs toward improved model parameterization and model structure for better model fidelity.

The results of the sensitivity analysis presented here can be utilized to construct the parameter-output response surface for the CLM4.5(ED) model, which can assist future efforts for model calibration or diagnosis. These findings may help us better understand the overall model structure and guide the estimation of key model parameters with significant control over vegetative processes in these models for better model fitting to data. The FAST analysis provides a promising means of analyzing complex LSM components, and can be a powerful tool in understanding the necessarily high-dimensional representation of living

systems within Earth System models.

*Code and data availability.* To access the CLM4.5(ED) source code, visit github.com/NGEET/fates-release. The FAST methodology described herein is available at sites.google.com/site/xuchongang/uasatoolbox. The version of the model codes used in this paper and the corresponding model simulations from all 5000 parameter combinations as well as simulation of the default parameter set are available at NGEE tropic data archive (https://ngt-data.lbl.gov, data id: NGT0091) and also upon request from the corresponding authors.




## Appendix A: Allometry equations

The following equations are cohort-based calculations for allometry in CLM4.5(ED). Interested readers are referred to *Fisher et al.* (2015) for more information. The parameters used for the allometry equations include $dbh2h_\mathrm{m}$, $dbh2h_\mathrm{c}$, $dbh2bd_\mathrm{a}$, $dbh2bd_\mathrm{b}$, $dbh2bd_\mathrm{c}$, and $dbh2bd_\mathrm{d}$ (all are unitless variables). Specifically, the dead wood biomass (BD; Kg C) is calculated as a function of diameter (DBH; cm), height (h; meter) and wood density (g cm$^{-3}$),

$$BD = (dbh2bd_\mathrm{a})(h^{dbh2bd_\mathrm{b}})(DBH^{dbh2bd_\mathrm{c}})(density_\mathrm{wood}^{dbh2bd_\mathrm{d}}) \tag{A1}$$

The height (m) is calculated based on $DBH$ (cm) as follows:

$$H = 10^{dbh2h_\mathrm{c}}(DBH^{dbh2bd_\mathrm{m}}) \tag{A2}$$

## Appendix B: Temperature response curve

The parameters used for the temperature response curve equations include the equation to calculate the maximum carboxylation rate, $V_\mathrm{c,max25}$, the maximum electron transport rate, $J_\mathrm{max}$, and the Triose phosphate use (TPU) limited carboxylation rate, $TPU$ (also all parameters here are unitless) (*Fisher et al.*, 2015). The temperature response equations for $V_\mathrm{c,max,z}$, $J_\mathrm{max,z}$, and $TPU\mathrm{z}$ are:

$$V_\mathrm{c,max,z} = V_\mathrm{c,max,25}(e^{\frac{vcmaxha}{(0.001rgas)(t_\mathrm{frz}+25)}})(1 - \frac{t_\mathrm{frz}+25}{t_\mathrm{veg}})(\frac{vcmaxc}{1 + e^{-vcmaxhd+(vcmaxse)(t_\mathrm{veg})}}) \tag{B1}$$

$$J_\mathrm{max,z} = J_\mathrm{max,25}(e^{\frac{jmaxha}{(0.001rgas)(t_\mathrm{frz}+25)}})(1 - \frac{t_\mathrm{frz}+25}{t_\mathrm{veg}})(\frac{jmaxc}{1 + e^{-jmaxhd+(jmaxse)(t_\mathrm{veg})}}) \tag{B2}$$

$$TPU_\mathrm{z} = tpu25(e^{\frac{tpuha}{(0.001rgas)(t_\mathrm{frz}+25)}})(1 - \frac{t_\mathrm{frz}+25}{t_\mathrm{veg}})(\frac{tpuc}{1 + e^{-tpuhd+(tpuse)(t_\mathrm{veg})}}) \tag{B3}$$

where $t_\mathrm{frz}$ is the freezing point of water in Kelvin (273.15 K).



### Appendix C: Carbon storage in CLM4.5(ED)

The target carbon storage is the *cushion* parameter shown in Table D3. Specifically, a higher value of this parameter will lead to a higher allocation of carbon to storage and thus lower allocation to growth at the specific time step. Also, carbon storage plays an important role for the simulated mortality through the parameter that controls the mortality rate under stress,

*stress_mort* in Table D3. The tree will be under stress when it has low carbon storage (< leaf biomass). Therefore the target carbon storage parameter and the mortality rate under stress parameter play a large role in determining the level of mortality that occurs in the simulations.

Carbon storage, $b_{store}$ (in kg C/cohort) plays a very important role in both growth and mortality (*Fisher et al.*, 2015). Specifically, CLM4.5(ED) assumes a target carbon storage determined by the multiplication of leaf biomass ($b_{leaf}$) and the target

carbon storage parameter (i.e. the target amount of carbon plants store relative to leaf biomass; $S_{cushion}$, variable *cushion* in Table D3). At the specific time, the carbon balance for growth and storage is calculated as follows,

$$C = NPP - T_{md}f_{md,min} \qquad (C1)$$

where $T_{md}$ is the maintenance respiration and $f_{md,min}$ is the minimum fraction of the maintenance demand (storage priority parameter in Table D1) that the plant must meet each time step, which represents a life-history-strategy decision concerning

whether leaves should remain on in the case of low carbon uptake (a risky strategy) or not be replaced (a conservative strategy).

The fraction of the carbon balance for each cohort allocated to the carbon storage pool ($f_{store}$) will be determined by the following equations:

$$f_{store} = e^{(-f_{tstore})^4} \qquad (C2)$$

where

$$f_{tstore} = \max\left(0, \frac{b_{store}}{S_{cushion}b_{leaf}}\right) \qquad (C3)$$

Thus, the target carbon storage parameter, $S_{cushion}$, can affect carbon allocations. Specifically, a higher value of $S_{cushion}$ will lead to a higher allocation of carbon to storage and thus lower allocation to growth at the specific time step.

Carbon storage also plays an important role for the mortality. Specifically, carbon starvation mortality ($M_{cs}$) is calculated as follows:

$$M_{cs} = S_{m}\max\left(0, 1 - \frac{b_{store}}{b_{leaf}}\right) \qquad (C4)$$

where $S_{m}$ is the stress mortality factor (i.e., *stress_mort* in Table D3).





## Appendix D: Appendix Figures and Tables

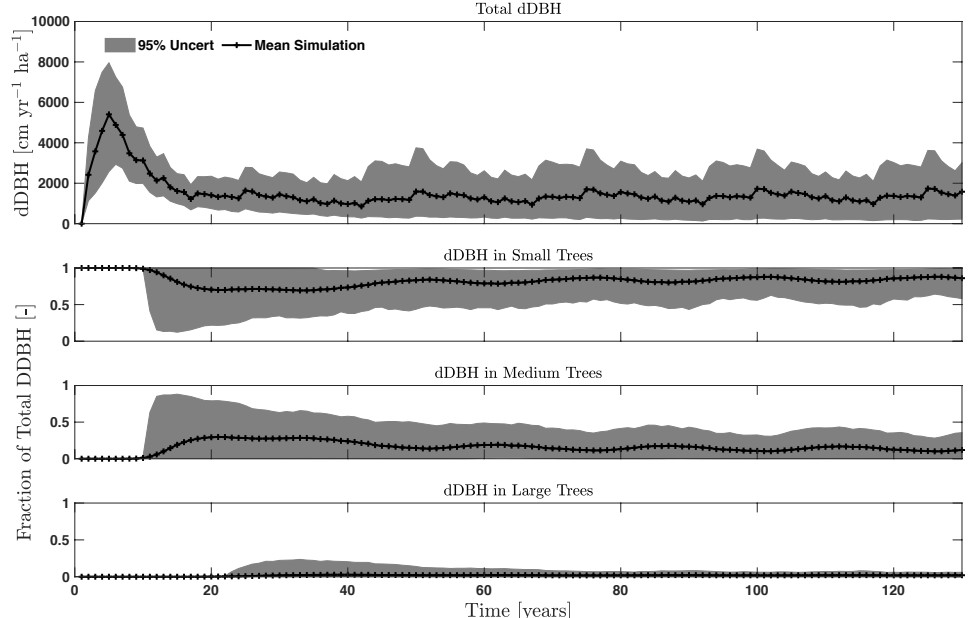

**Figure D1.** Simulated total change in diameter at breast height (dDBH) from CLM4.5(ED) for all trees and its fractional distribution for small ($diameter < 10$ cm), medium (10 cm $< diameter < 50$ cm), and large trees ($diameter > 50$ cm). Shown are the mean simulation (black line) with 95 % spread of the simulation ensemble.





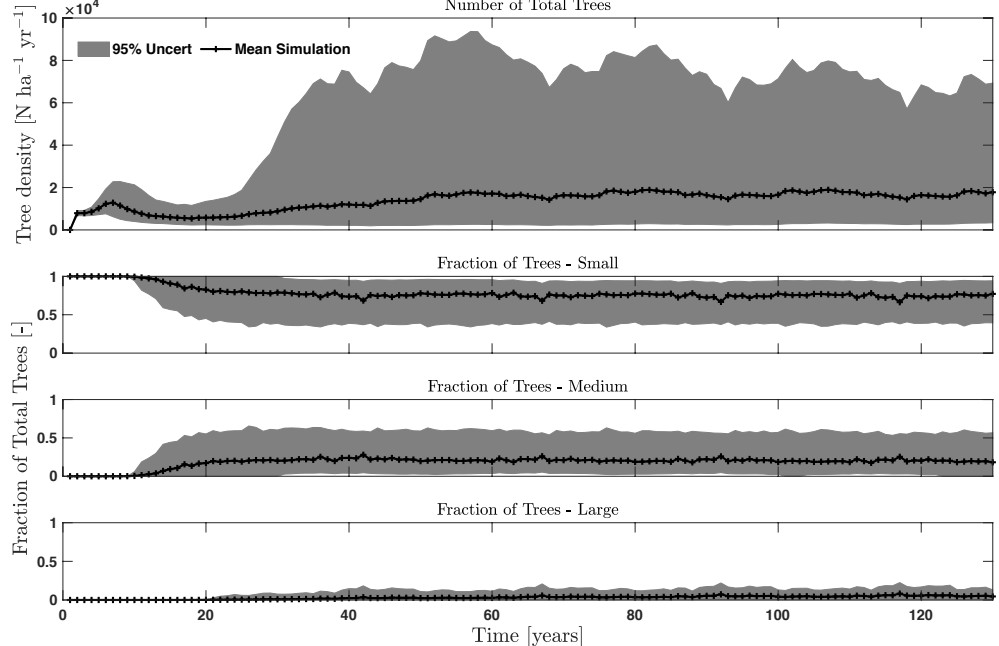

**Figure D2.** Simulated total number of trees per area (NPLANT) from CLM4.5(ED) for all trees and its fractional distribution for the various tree sizes considered, including small ($diameter < 10$ cm), medium ($10$ cm $< diameter < 50$ cm), and large trees ($diameter > 50$ cm). Shown are the mean simulation (black line) with 95 % spread of the simulation ensemble.



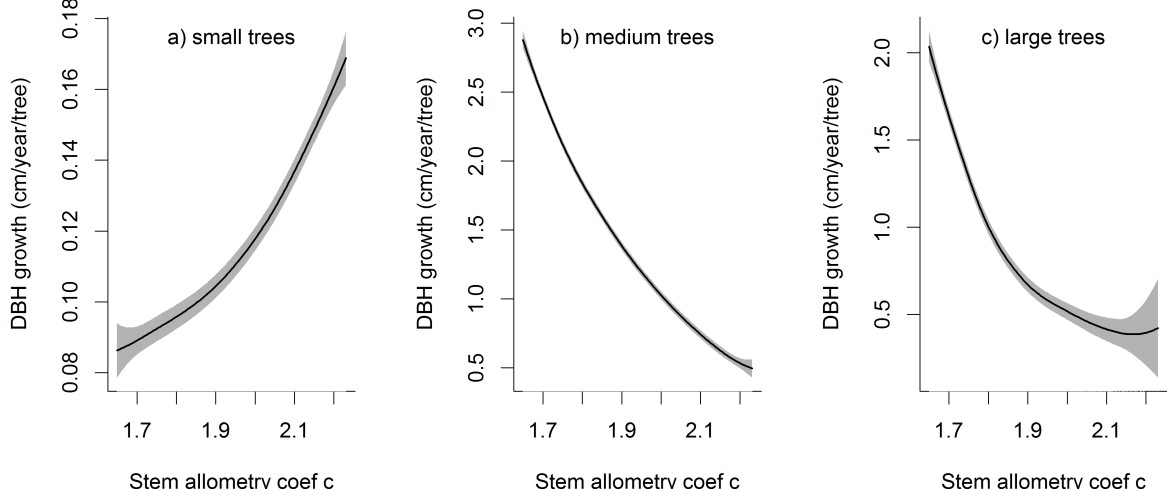

**Figure D3.** Impacts of stem allometry on the change in diameter at breast height (dDBH) averaged over the simulations years 100-130 for trees of different sizes. The shaded area shows the 95% confidence interval of these relations.





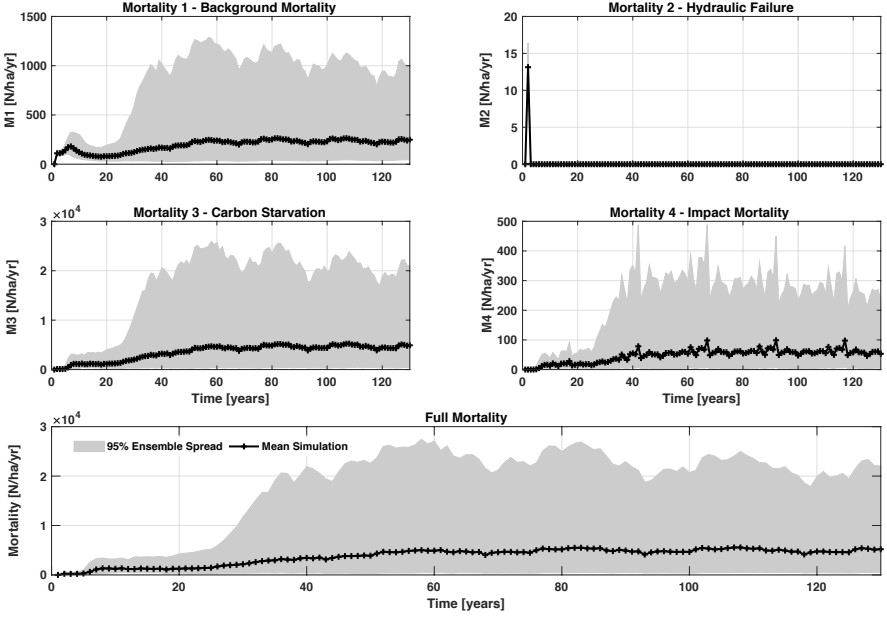

**Figure D4.** Mortality outputs from CLM4.5(ED), including the mechanisms of M1 - Background Mortality, M2 - Hydraulic Failure, M3 - Carbon Starvation, and M4 - Impact Mortality. The bottom panel shows the total mortality, which is the sum of M1-M4. An additional possibility for mortality in CLM4.5(ED) is from fire disturbances, however the fire sub-routine of the model is turned off since the study site is in the Amazon. Shown are 95 % (light grey) and 90 % (dark grey) spread of the simulation ensemble, along with the mean simulation (black lines) and the simulation using the default parameter set (green lines).





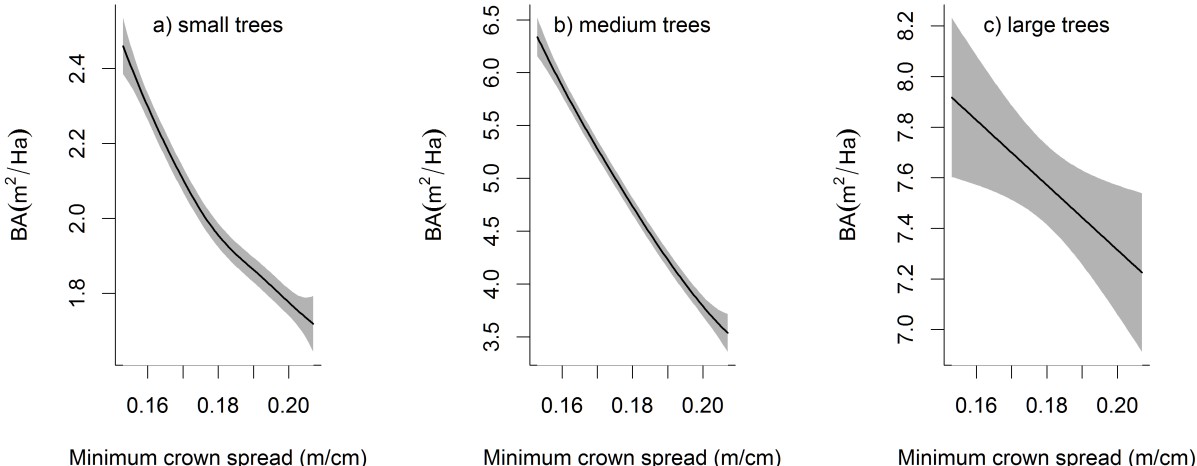

**Figure D5.** Impacts of minimum crown spread on the basal area (BA) averaged over the simulations years 100-130 for trees of different sizes. The shaded area shows the 95% confidence interval of these relations.





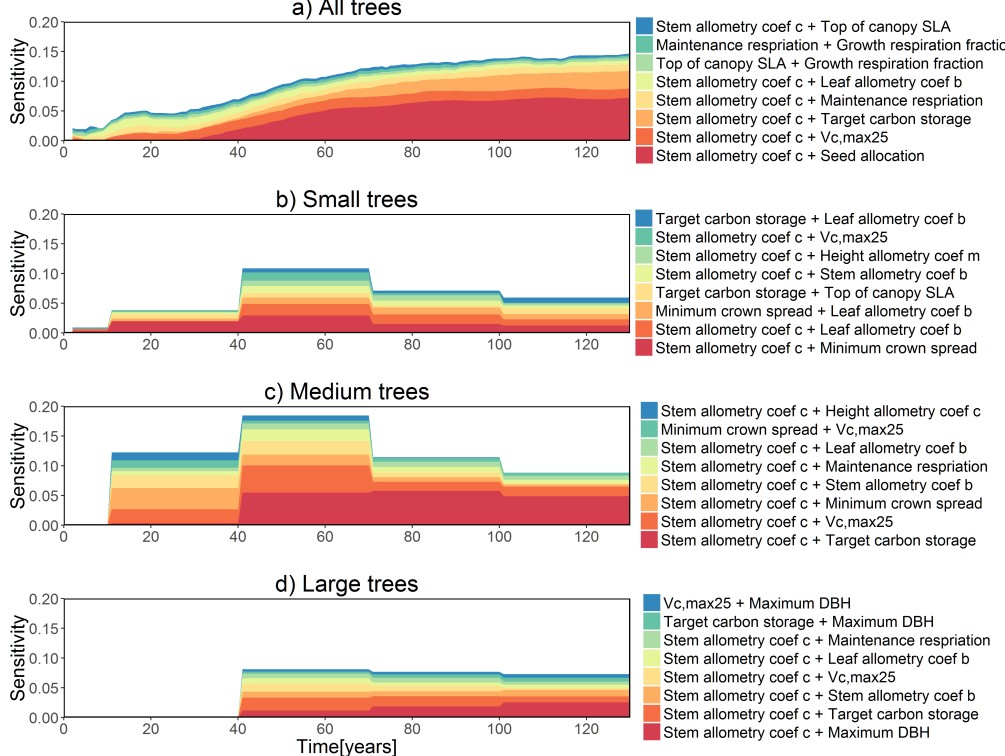

**Figure D6.** 2$^{nd}$ order sensitivity index of the model parameters for the Basal Area (BA) outputs from CLM4.5(ED) for all trees, small trees, medium trees and large trees. Shown are the top 8 most important parameter interactions, in order of importance (red is the most important and blue is the least important)





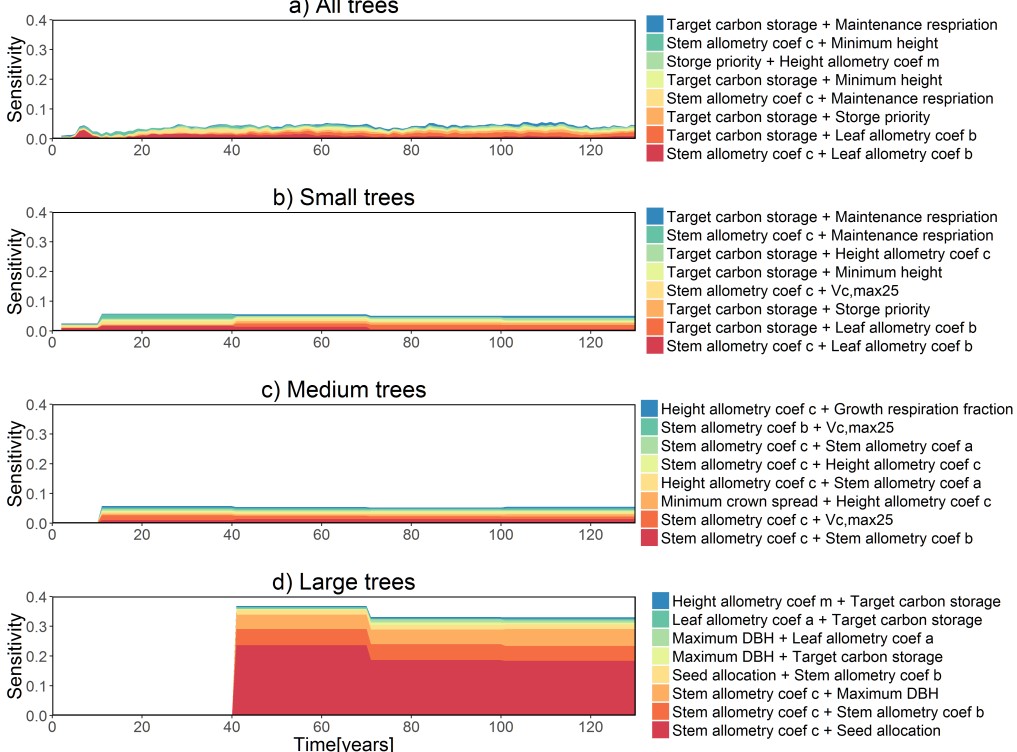

**Figure D7.** 2$^{nd}$ order sensitivity index of the model parameters for the change in diameter at breast height (dDBH) outputs from CLM4.5(ED) for all trees, small trees, medium trees and large trees.Shown are the top 8 most important parameter interactions, in order of importance (red is the most important and blue is the least important)





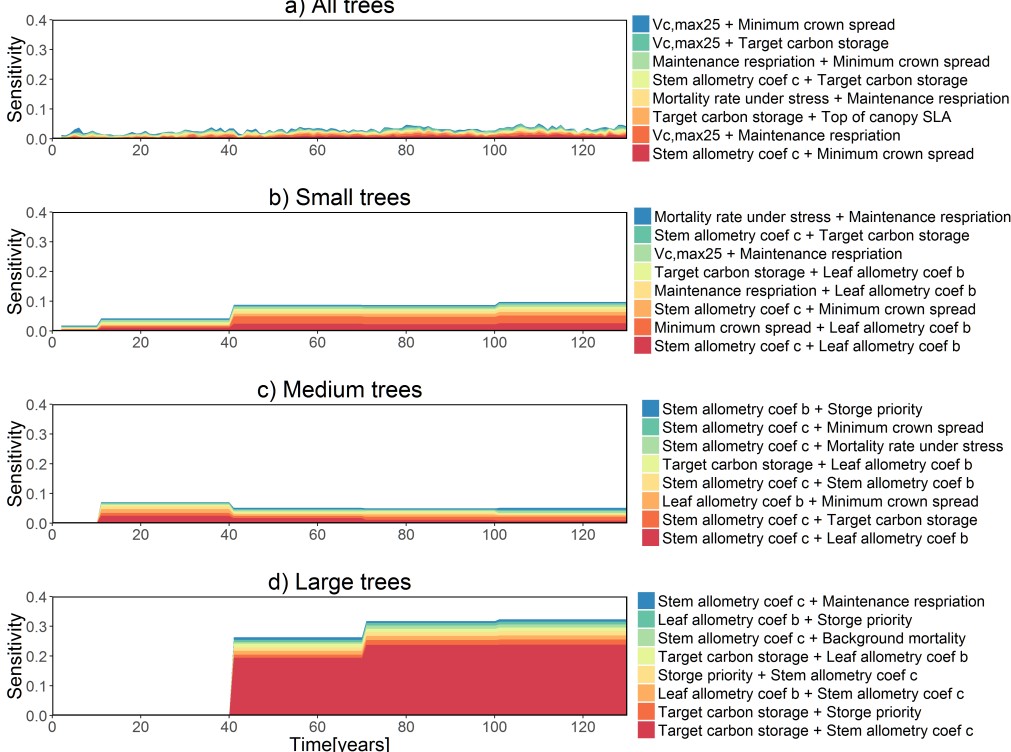

**Figure D8.** 2nd order sensitivity index of the model parameters for the mortality outputs from CLM4.5(ED) for all trees, small trees, medium trees and large trees.Shown are the top 8 most important parameter interactions, in order of importance (red is the most important and blue is the least important)





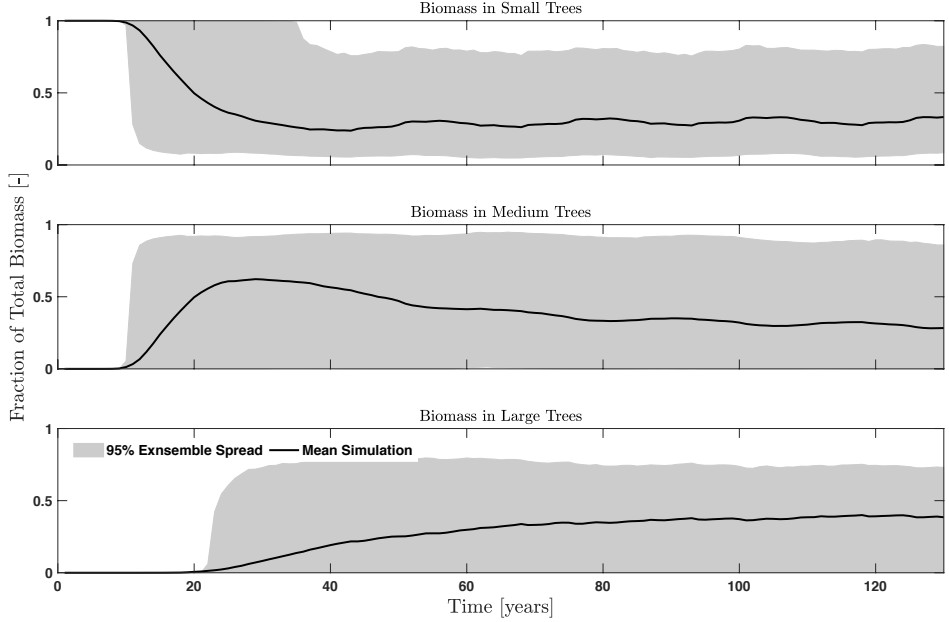

**Figure D9.** Fraction of total biomass for trees of different sizes, including small ($diameter < 10$ cm), medium ($10$ cm $< diameter < 50$ cm), and large trees ($diameter > 50$ cm).





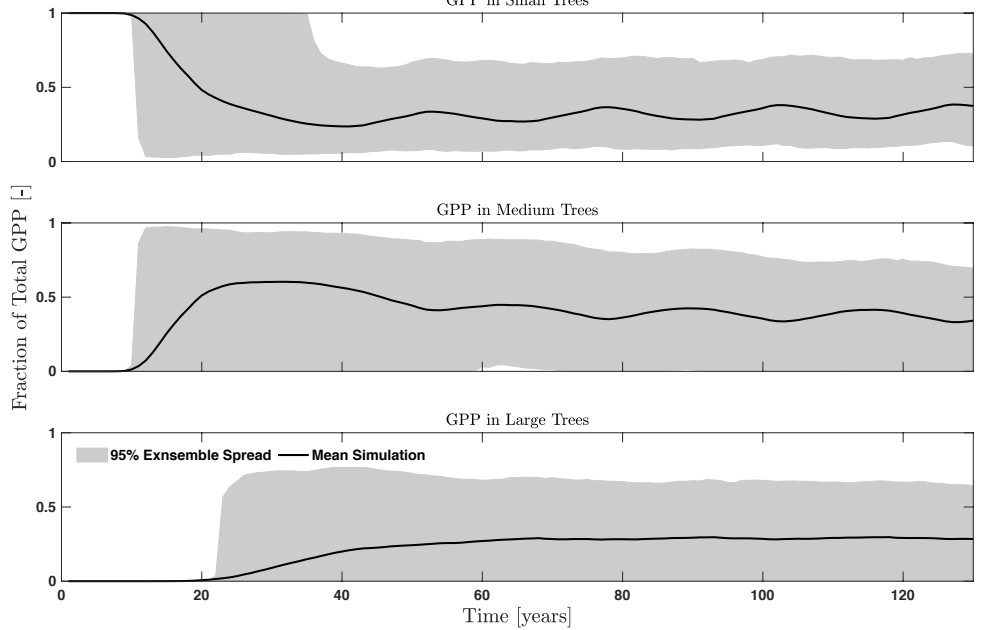

**Figure D10.** Fraction of total GPP for trees of different sizes, including small ($diameter < 10$ cm), medium ($10$ cm $< diameter < 50$ cm), and large trees ($diameter > 50$ cm).



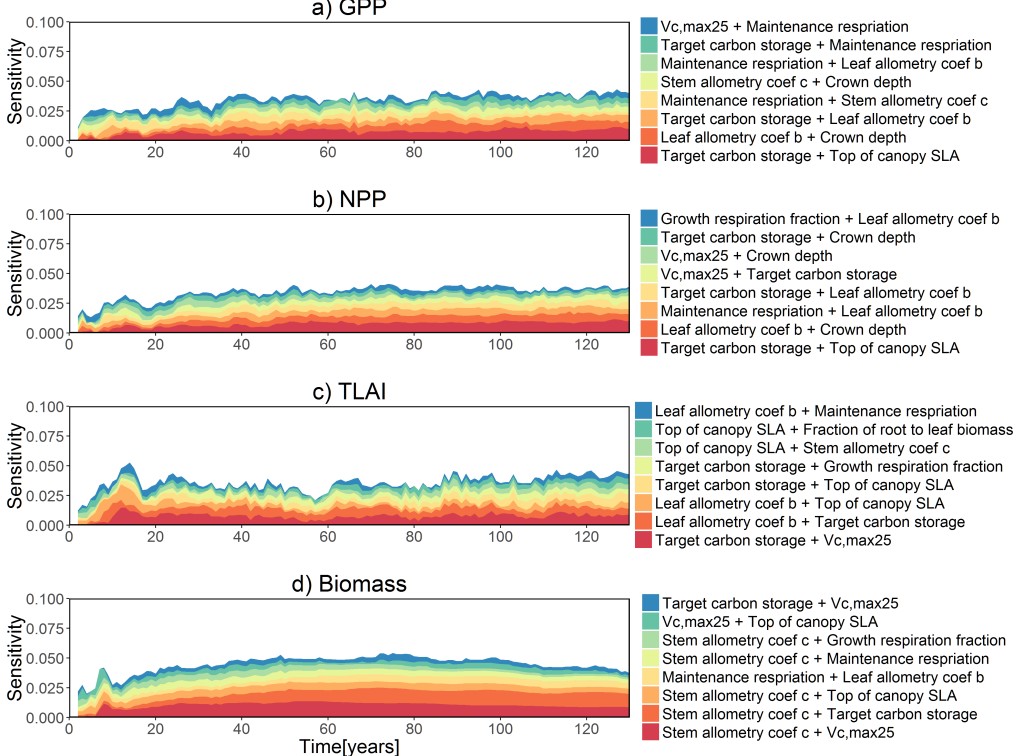

**Figure D11.** 2[nd] order sensitivity index of the model parameters for the GPP, NPP, LAI, and biomass outputs from CLM4.5(ED). Shown are the top 8 most important parameter interactions, in order of importance (red is the most important and blue is the least important)





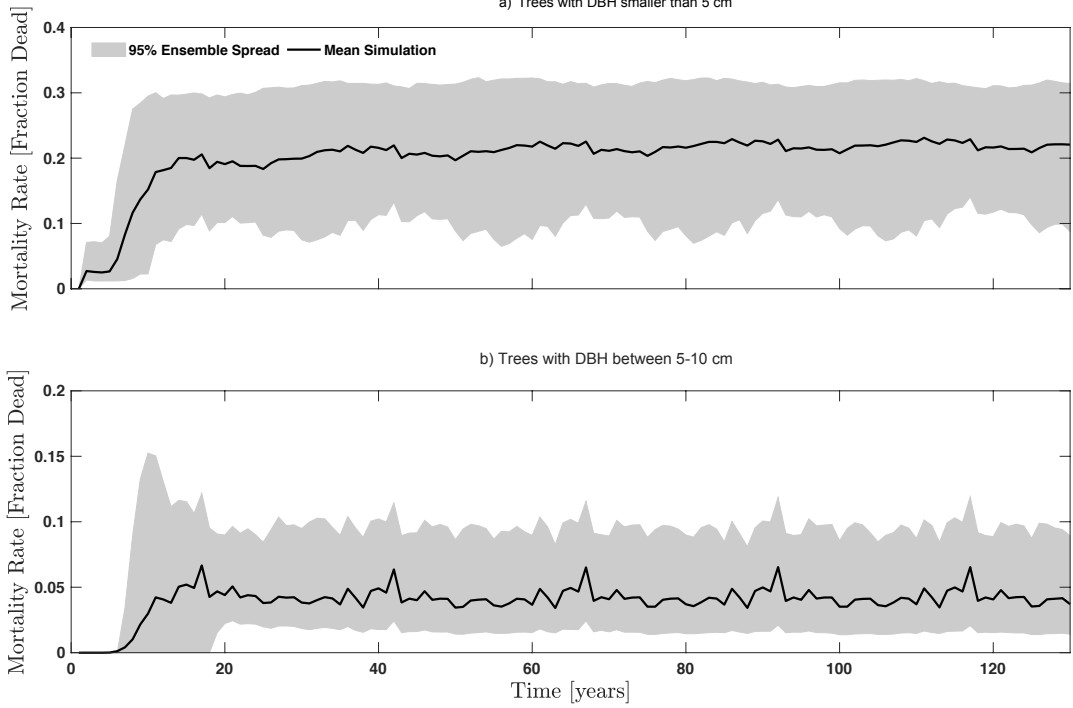

**Figure D12.** Mortality outputs from CLM4.5(ED) for trees with DBH smaller than 5 cm (Panel A) and all trees with DBH between 5 and 10 cm (Panel B). Shown are 95 % (light grey) spread of the simulation ensemble, along with the mean simulation (black lines).





**Table D1.** Parameter sets used in this study - Part 1

| Name | Variable Name | Units | Default | Lower | Upper |
|------|---------------|-------|---------|-------|-------|
| **Allocation and Allometry Parameters** | | | | | |
| Height allometry coef m | $dbh2h_m$ | [-] | 0.64 | 0.54 | 0.74 |
| Height allometry coef c | $dbh2h_c$ | [-] | 0.37 | 0.31 | 0.43 |
| Leaf allometry coef a | $dbh2bl_a$ | [-] | 0.042 | 0.036 | 0.048 |
| Leaf allometry coef b | $dbh2bl_b$ | [-] | 1.56 | 1.33 | 1.79 |
| Leaf allometry coef c | $dbh2bl_c$ | [-] | 0.55 | 0.47 | 0.63 |
| Leaf allometry SLA scaler | $dbh2bl\_slascaler$ | [-] | 0.03 | 0.025 | 0.035 |
| Stem allometry coef a | $dbh2bd_a$ | [-] | 0.069 | 0.059 | 0.079 |
| Stem allometry coef b | $dbh2bd_b$ | [-] | 0.57 | 0.49 | 0.66 |
| Stem allometry coef c | $dbh2bd_c$ | [-] | 1.94 | 1.65 | 2.23 |
| Stem allometry coef d | $dbh2bd_d$ | [-] | 0.93 | 0.79 | 1.07 |
| SAI scaler | $sai-scaler$ | [-] | 0.05 | 0.043 | 0.058 |
| Ratio of sapwood to leaf area | $sapwood-ratio$ | [$m^{-1}$] | 0.001 | 0.00085 | 0.00115 |
| Fraction of root to leaf biomass | $froot\_leaf$ | [gC/gC] | 1 | 0.85 | 1.15 |
| Seed allocation | $seed\_alloc$ | [0-1] | 0.1 | 0.085 | 0.115 |
| Fraction of aboveground stem | $ag\_biomass$ | [0-1] | 0.6 | 0.51 | 0.69 |
| Crown depth | $crown$ | [0-1] | 0.5 | 0.43 | 0.58 |
| Maximum crown spread | $maxspread$ | [cm/$m^2$] | 0.3 | 0.25 | 0.35 |
| Minimum crown spread | $minspread$ | [cm/$m^2$] | 0.18 | 0.15 | 0.21 |
| Root distribution coef a | $root_a$ | [$m^{-1}$] | 7 | 5.95 | 8.05 |
| Root distribution coef b | $root_b$ | [$m^{-1}$] | 1 | 0.85 | 1.15 |
| Maximum DBH | $max\_dbh$ | [cm] | 68 | 57.8 | 78.2 |
| Wood density | $wood\_density$ | [-] | 0.7 | 0.60 | 0.80 |
| Clone allocation | $clone\_alloc$ | [0-1] | 0.75 | 0.64 | 0.86 |



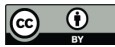

**Table D2.** Parameter sets used in this study - Part 2

| Name | Variable Name | Units | Default | Lower | Upper |
|---|---|---|---|---|---|
| **Regrowth Parameters** | | | | | |
| Initial seedling density | $initd$ | [m$^{-2}$] | 0.8 | 0.68 | 0.92 |
| Seed rain | $seed\_rain$ | [kgC/m$^2$/year] | 0.28 | 0.24 | 0.32 |
| Minimum height | $hgt\_min$ | [m] | 1.25 | 1.06 | 1.44 |
| **Photosynthetic and Respiration Parameters** | | | | | |
| Stomata conductance slope | $bb\_slope$ | [-] | 9 | 7.65 | 10.35 |
| $V_{c,max25}$ | $fnitr$ | [$umolCO_2 m^{-2}s^{-1}$)] | 60 | 51 | 69 |
| Leaf C:N | $leafcn$ | [gC/gN] | 30 | 25.5 | 34.5 |
| Storage priority | $leaf\_stor\_priority$ | [0-1] | 0.8 | 0.68 | 0.92 |
| Top of canopy SLA | $slatop$ | [m$^2$/gC] | 0.012 | 0.010 | 0.014 |
| Growth respiration fraction | $grperc$ | [-] | 0.3 | 0.26 | 0.34 |
| Maintenance respriation | $lmr25top$ | [$umolCO_2 m^{-2}s^{-1}$)] | 0.71 | 0.60 | 0.82 |
| Soil water potential for stomata closure | $smpsc$ | [mm] | -2.55e4 | -2.93e4 | -2.16e4 |
| Soil water potential for opening stomata | $smpso$ | [mm] | -6.60e4 | -7.59e4 | -5.61e4 |
| **Temperature Response Parameters** | | | | | |
| Vcmax temperature coef ha | $vcmaxha$ | [-] | 6.53e4 | 5.55e4 | 7.51e4 |
| Jmax temperature coef ha | $jmaxha$ | [-] | 4.35e4 | 3.70e4 | 5.00e4 |
| TPU temperature coef ha | $tpuha$ | [-] | 5.31e4 | 4.51e4 | 6.10e4 |
| Maintenance respiration coef ha | $lmrha$ | [-] | 4.63e4 | 3.94e4 | 5.33e4 |
| Vcmax temperature coef hd | $vcmaxhd$ | [-] | 14.92e4 | 12.68e4 | 17.16e4 |
| Jmax temperature coef hd | $jmaxhd$ | [-] | 15.20e4 | 12.92e4 | 17.48e4 |
| TPU temperature coef hd | $tpuhd$ | [-] | 15.06e4 | 12.80e4 | 17.32e4 |
| Maintenance respiration coef hd | $lmrhd$ | [-] | 15.06e4 | 12.80e4 | 17.32e4 |
| Vcmax temperature coef se | $vcmaxse$ | [-] | 485 | 412 | 558 |
| Jmax temperature coef se | $jmaxse$ | [-] | 495 | 420 | 570 |
| TPU temperature coef se | $tpuse$ | [-] | 490 | 416 | 564 |
| Maintenance respiration coef se | $lmrse$ | [-] | 490 | 416 | 564 |





**Table D3.** Parameter sets used in this study - Part 3

| Name | Variable Name | Units | Default | Lower | Upper |
|------|---------------|-------|---------|-------|-------|
| **Mortality Parameters** | | | | | |
| Background mortality | $b\_mort$ | [ yr$^{-1}$ ] | 0.014 | 0.012 | 0.016 |
| Target carbon storage | $cushion$ | ratio of leaf biomass | 1.2 | 1.02 | 1.38 |
| Mortality rate under stress | $stress\_mort$ | [ yr$^{-1}$ ] | 0.6 | 0.51 | 0.69 |
| Understory mortality rate | $understorey\_death$ | [-] | 0.56 | 0.48 | 0.64 |
| Seed mortality rate | $sd\_mort$ | [ yr$^{-1}$ ] | 0.98 | 0.83 | 1.0 |
| Hydraulic failure threshold | $hf\_sm\_threshold$ | [-] | 1.00e-6 | 8.5e-7 | 1.15e-6 |
| **Turnover Parameters** | | | | | |
| Leaf longevity | $leaf\_long$ | [years] | 1.5 | 1.28 | 1.72 |
| Root longevity | $root\_long$ | [years] | 1 | 0.85 | 1.15 |
| Stem Turnover | $alpha\_stem$ | [years] | 0.01 | 0.0085 | 0.0115 |
| **Radiation Parameters** | | | | | |
| Leaf reflectance: near-IR | $rholnir$ | [0-1] | 0.45 | 0.38 | 0.52 |
| Leaf reflectance: visible | $rholvis$ | [0-1] | 0.1 | 0.085 | 0.115 |
| Stem reflectance: near-IR | $rhosnir$ | [0-1] | 0.39 | 0.33 | 0.45 |
| Stem reflectance: visible | $rhosvis$ | [0-1] | 0.16 | 0.14 | 0.18 |
| Leaf transmittance: near-IR | $taulnir$ | [0-1] | 0.25 | 0.21 | 0.29 |
| Leaf transmittance: visible | $taulvis$ | [0-1] | 0.05 | 0.043 | 0.058 |
| Stem transmittance: near-IR | $tausnir$ | [0-1] | 1.00e-3 | 8.5e-4 | 1.15e-3 |
| Stem transmittance: visible | $tausvis$ | [0-1] | 1.00e-3 | 8.5e-4 | 1.15e-3 |
| Leaf orientation index | $xl$ | $[-0.4 < xl < 0.6]$ | 0.1 | 0.085 | 0.115 |
| **Competition Parameters** | | | | | |
| Competitive exclusion parameter | $comp\_excln$ | [-] | 0.1 | 0.085 | 0.115 |





**Table D4.** Parameter sets used in this study - Part 4

| Name | Variable Name | Units | Default | Lower | Upper |
|------|---------------|-------|---------|-------|-------|
| **Phenology Parameters** | | | | | |
| Drought deciduous threshold | $ed\_ph_{\text{drought-threshold}}$ | [0-1] | 0.15 | 0.13 | 0.17 |
| Phenology coef a | $ed\_ph_{\text{a}}$ | [-] | -68 | -78.2 | -57.8 |
| Phenology coef b | $ed\_ph_{\text{b}}$ | [-] | 638 | 542.3 | 733.7 |
| Phenology coef c | $ed\_ph_{\text{c}}$ | [-] | -1.00e-3 | -1.15e-3 | -8.5e-4 |
| Chilling day temperature | $ed\_ph_{\text{chiltemp}}$ | °C | 5 | 4.25 | 5.75 |
| Cold day temperature | $ed\_ph_{\text{coldtemp}}$ | °C | 7.5 | 6.4 | 8.6 |
| Cold days for leave drop off | $ed\_ph_{\text{ncolddayslim}}$ | days | 5 | 4.3 | 5.8 |
| Minimum days before leaf on | $ed\_ph_{\text{mindayson}}$ | days | 30 | 25 | 35 |
| Minimum days before leaf drops | $ed\_ph_{\text{doff-time}}$ | days | 100 | 85 | 115 |
| Seed turnover | $seed\_turnover$ | [ yr$^{-1}$ ] | 0.51 | 0.43 | 0.59 |
| Germination rate | $germination\_timescale$ | [ yr$^{-1}$] | 0.5 | 0.43 | 0.58 |
| **Aerodynamic Parameters** | | | | | |
| Leaf dimension | $dleaf$ | [m] | 0.04 | 0.034 | 0.046 |
| Momentum roughness length | $z0mr$ | [-] | 0.075 | 0.064 | 0.086 |
| Displacement height ratio | $displar$ | m | 0.67 | 0.57 | 0.77 |
| **Additional Parameters** | | | | | |
| Freeze tolerant temperature | $freezetol$ | [-] | 1000 | 850 | 1150 |
| Respiration response factor to drought | $resp\_drought\_response$ | [-] | 0.5 | 0.43 | 0.58 |
| Soil moisture factor for growth | $soilbeta$ | [-] | 2000 | 1700 | 2300 |
| Maximum leaf water potential | $leafwatermax$ | [-] | 0.1 | 0.085 | 0.115 |
| Root water resistance | $rootresist$ | [-] | 200 | 170 | 230 |
| Dispersal distance | $dispersal$ | [-] | 0.5 | 0.43 | 0.58 |
| Cohort fuse tolerance | $profile\_tol$ | [-] | 0.7 | 0.60 | 0.80 |





*Author contributions.* All authors contributed to the manuscript writing. CX designed the numerical experiments, developed scripts for sensitivity analysis, and analyzed model results; EM implemented the model runing, extracted the model outputs and analyzed model results; RF, RK, CK, CX and BC contributed to the model development and simulations; JH, DR,SS and AW provide suggestions on senstivity analysis and LW provided support for model simulations; DJ provide data on model evaluations; NM, LK, JC and JV provide support and guidance on the experiment and manuscript.

*Competing interests.* We declare that no competing interests are present in the manuscript.

*Acknowledgements.* This work was supported by a United States Department of Energy (US DOE) Office of Science Next Generation Ecosystem Experiment at Tropics (NGEE-T) project and the Graduate Student Researcher (SCGSR) fellowship. EM and JV are also supported by funding from the UC-Lab Fees Research Program Award 237285. Model simulations were made possible thanks to the Conejo Super Computing system at the Los Alamos National Laboratory (LANL).

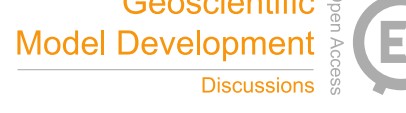

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
