# Peer review of "Identification of key parameters controlling demographically structured vegetation dynamics in a Land Surface Model [CLM4.5(FATES)]"

_Geoscientific Model Development, 2019_

## Short Comment (SC1) · 13 Feb 2019

In this study, the authors used a type of sensitivity analyses method to understand the behavior of a developed land-surface model (CLM4.5(ED)) to changes in parameter values. They arrive at results which other land surface models such as those that focus more on bio-geochemistry (e.g. CLM4.5) or those that focus more on vegetation dynamics (e.g. ED) could have also come up with if these models were ran separately. While their sensitivity analyses method is okay, their results are trivial. I have concerns regarding the actual simulation method they have used in this study.

My concerns are on the bias and uncertainties the authors may have in their results. I

list the points below, which are likely to be interconnected.

1) The entire parameter space of the model was not explored so how can this be even called a global sensitivity analyses? You looked at 87 parameters in this study. How much uncertainty you have in your existing results for the parameters that you have ignored?

2) How does these results of CLM4.5(ED) compare with the other versions of CLM e.g. with CLM-DGVM or CLM-FATES?

3) The model simulations are performed for 1 deg x 1 deg (approximately at 100 km). This resolution is quite coarse. If you are trying to understand the large-scale vegetation responses to changes in parameter values, then I think that needs to be made clear (at-least in the abstract as well as in the introduction). If not, then you need to address how much your results will change if you did the sensitivity analyses at the local scale using local weather conditions.

4) This simulation is only carried out at one site. Why was this specific site chosen? Isn't this already a bias? Will you get similar results at other biomes?

5) The climate data was recycled, which might be okay, but you used climate data from 1942 to 1972? I don't think you can compare your modeled results with observations unless you believe that the climate at your studied site didn't change much or if your measurements were carried out around 1972? Further, isn't CLM4.5(ED) sensitive to climate forcing?

6) The simulation was carried out for about 130 years, where the changes in parameter values (+/- 15%) was relatively small compared to the default value. This % change was fixed for all parameters. Isn't there any parameter out of 87 that has a wider range in reality. If so, how can one be really sure about these results then?

7) The authors should quantify the relative impacts on the carbon fluxes or vegetation stocks due to parameter changes, and state whether these impacts are statistically

significant or not. At present, it is unclear how much the identified parameters control the carbon fluxes or stocks.

---

## Short Comment (SC2) · 15 Feb 2019

Vick,

Thank you very much for the comments. Please see below a short response before our final revision.

1) The entire parameter space of the model was not explored so how can this be even called a global sensitivity analyses? You looked at 87 parameters in this study. How much uncertainty you have in your existing results for the parameters that you have ignored?

[Figure]

Response: There are two types of analysis for models. One is the sensitivity analysis, which is used to explore the sensitivity of model outputs to parameter changes. They normally change the parameter by an equal amount/percentage to understand the model behaviors. A second type is the uncertainty analysis, which is used to understand how much uncertainty or variability is in the model outputs and what contributes to the uncertainty. It is possible that an output is very sensitive to parameters but has less uncertainty contributions if we have a good estimate of the parameter. Both analyses will be useful for model development with the sensitivity analysis focusing on understanding the baseline of model behaviors and the uncertainty analysis focusing on guiding field and laboratory measurements.

Our study is the sensitivity analysis. The "global" here is refer to that we change all the parameters at the simultaneously for understanding the impact of parameter of model outputs. There is confusion on how we define "global" sensitivity analysis. A sensitivity analysis is considered to be global when all the input factors are varied simultaneously and the sensitivity is evaluated over their entire range of interest (https://link.springer.com/referenceworkentry/10.1007%2F978-0-387-35973-1_538). From methodological perspective, it is mainly that we change the parameters simultaneously. From the scientific perspective, the question is what the entire range of interests is. For sensitivity analysis studies, the entire range of interest could be a certain percentage of default values of parameters. For uncertainty analysis, the entire range of interest could be the uncertainty or ranges in the measurements and observations.

2) How does these results of CLM4.5(ED) compare with the other versions of CLM e.g. with CLM-DGVM or CLM-FATES?

Response: This version of model [CLM4.5(ED)] is the initial version of CLM-FATES. We changed the name from ED to FATES about 2 years ago. We will make this clear for the revision to avoid confusions. As far as we know, we have no studies for CLM-CNDV, which is an original version of the dynamic vegetation for CLM. We did compare
our results with ED in the manuscript.

3) The model simulations are performed for 1 deg x 1 deg (approximately at 100 km). This resolution is quite coarse. If you are trying to understand the large-scale vegetation responses to changes in parameter values, then I think that needs to be made clear (at-least in the abstract as well as in the introduction). If not, then you need to address how much your results will change if you did the sensitivity analyses at the local scale using local weather conditions.

Response: We will make this clear for our revised manuscripts.

4) This simulation is only carried out at one site. Why was this specific site chosen? Isn't this already a bias? Will you get similar results at other biomes?

Response: This a good question. We chose this site because CLM(ED) is already set up for this site and is common test site for the tropical biome. We will expect to see different results for other biomes but we will expect the main results will maintain valid. This is our first sensitivity of the model and we will see other research groups working on different sites to improve our understanding of the model at different locations.

5) The climate data was recycled, which might be okay, but you used climate data from 1942 to 1972? I don't think you can compare your modeled results with observations unless you believe that the climate at your studied site didn't change much or if your measurements were carried out around 1972? Further, isn't CLM4.5(ED) sensitive to climate forcing?

Response: Yes, CLM4.5 is sensitive to climate forcing. We understand your concern about the climate driver and I agree with you of the potential bias. However, because we are more on the qualitative comparison using different data at different periods of times, we feel that the bias could be small.

6) The simulation was carried out for about 130 years, where the changes in parameter values (+/- 15%) was relatively small compared to the default value. This % change

was fixed for all parameters. Isn't there any parameter out of 87 that has a wider range in reality. If so, how can one be really sure about these results then?

Response: As I pointed out at beginning of response, this study is a sensitivity analysis focusing understanding of the model behaviors. Ongoing studies of uncertainty analysis will help us understand the uncertainty contributions.

7) The authors should quantify the relative impacts on the carbon fluxes or vegetation stocks due to parameter changes, and state whether these impacts are statistically significant or not. At present, it is unclear how much the identified parameters control the carbon fluxes or stocks.

Response: We do have the standard errors of estimated sensitivities based on the delta method. See details from Chonggang Xu & George Zdzislaw Gertner (2011) Reliability of global sensitivity indices, Journal of Statistical Computation and Simulation,81:12, 1939-1969, DOI: 10.1080/00949655.2010.509317. We will update the p-values for the revision. We do plots the proportion of contributions in the sensitivity figures.

Yours Chonggang

―――――――――――――――――――――――

---

## Referee Comment (RC1) · Nancy Kiang (Referee) · 22 Feb 2019

Identification of key parameters controlling demographicallystructured vegetation dynamics in a Land Surface Model [CLM4.5(ED)] Elias C. Massoud et al.

This paper uses the Fourier Amplitude Sensitivity Test (FAST) method to perform a parameter sensitivity study for the CLM4.5(ED) model. The study simulates the variance range and first and second order sensitivities for particular diagnostics relative to parameter perturbations drawn from uniform sampling within $\pm15\%$ of the model default values for 87 parameters, including biophysical (including temperate response), allometric, allocation, reproduction and establishment, mortality, leaf optical, leaf longevity

parameters, and a "competitive exclusion" parameter. The study is performed for one PFT at an Amazon forest site with 25 years of recycled meteorological forcing at 1°x1°, with 5000 simulations (a little less than the ~100 x number of parameters called for by Xu and Germer, 2011, so it seems ~37 of the 87 parameters were not "important") 130 years in length.

This study serves more as a template and foundation for further work to perform later parameter optimization and more rigorous analysis relative to data, so as such it provides good documentation of methods of setup and analysis. The authors largely acknowledge typical concerns about the shortcomings of sensitivity studies like this, including that the parameter sampling is not based on observed distributions, only one PFT is simulated, and the mortality sensitivity to hydraulic failure is not a model sensitivity but rather a site meteorological forcing result. The choice of this particular site was obviously just expedient based on available drivers, so this reviewer views the study more as preliminary setup and test of concept, rather than new findings about DGVMs or about nature.

While the results are largely confirming what is already known about the model, such as the overshading of understory trees with the PPA, and the non-linearity of responses by tree size (and some of the results could have been obtained analytically from the equations in the DGVM), the technique illustrates a method quantitatively to stratify or rank the sensitivity of a diagnostic by parameters in a way not available through just Monte Carlo sampling.

For the sake of a sensitivity analysis method, the authors should add more to the discussion about the following:

1) Whether the 25-year periodicity of the meteorological forcings (very apparent in Figures 3c and 3d) affects the purported parameter sensitivity.

2) A little more explanation about the 30-year intervals chosen to average the sensitivity values. The authors say, "This is done in view that the transient and abrupt

changes across different size categories in the annual model outputs could make the FAST analysis only account for a minor amount of the variance contribution from each parameter." If the FAST sensitivities are a function of temporal averaging period, this is rather important to address! In that case, it seems running means and a spectral averaging approach would make sense to identify time scales of sensitivity. This perhaps is a missed opportunity to show something interesting in terms of the model sensitivity; e.g. there are model-dependent fast and slow processes updated at short time scales, some processes updated at longer time scales, as well as event–driven processes dependent on meteorological forcing.

3) For the sake of readers not familiar with FAST, how does the underlying distribution of parameters propagate to the calculation of the variance, the latter by definition assuming Gaussian distributions. Some sentences on this would be good to add, citing statistical theory papers, and not just application papers.

4) That the sensitivity values change with time and community structure, but the parameters do not change their relative rank to each other: is this a result of the model structure (one PFT, fixed sensitivities in the physics) or a result of the site meteorology?

5) A suggestion/question: Is it possible to do surface analysis of two-variable sensitivities, and would that reveal any useful relations?

Other small things to fix: Section 1 of the paper needs to be proofread for grammar and subject verb agreement. There are a lot of sentences that are a bit sloppy. After the Section 1, this problem disappears! Overall, the paper is clearly written, well-documented, figures illustrate results well. As the Holm et al. (2018, In Review) paper cited picks up where this paper leaves off with data on the parameter distributions to provide for some quantitative understanding of the system being profiled, this paper by Massoud et al as primarily a methodological exercise is fine for a journal like GMD.

---

## Referee Comment (RC2) · Xiangtao Xu (Referee) · 23 Feb 2019

General Comment:

This paper applies the Fourier Amplitude Sensitivity Test (FAST) to a land surface model that accounts for vegetation demography. In particular, the authors try to conduct a comprehensive assessment of the sensitivity in the simulated vegetation dynamics to over 80 input parameters that describe plant biochemical, allometric, and demographic traits. The analysis is performed for a tropical rainforest region in the Amazon, where model bears large uncertainty. Limited by data and computational cost, the paper only included one plant functional type and found that model results are very sensitive to

allometric parameters across all time-scales.

Generally, I feel the study is somewhat interesting for GMD in the sense that it introduces FAST to vegetation demography and ecosystem modeling, and that it includes allometric parameters. However, I think the paper can be improved in several aspects to be more useful to the community.

Scientific Comments

1. [P6L32-33] I understand the challenge to include trait-covariation in such analysis. However, the current assumption of absolute orthogonality between parameters makes it hard to interpret the results. For instance, Diaz et al. 2015 shows that the actual ecophysiologically-viable trait space might only be 2% of the total N-dimensional parameter space. It would be helpful to include some more discussions to help interpret the results

Díaz, S., Kattge, J., Cornelissen, J.H.C., Wright, I.J., Lavorel, S., Dray, S., Reu, B., Kleyer, M., Wirth, C., Colin Prentice, I., Garnier, E., Bönisch, G., Westoby, M., Poorter, H., Reich, P.B., Moles, A.T., Dickie, J., Gillison, A.N., Zanne, A.E., Chave, J., Joseph Wright, S., Sheremet'ev, S.N., Jactel, H., Baraloto, C., Cerabolini, B., Pierce, S., Shipley, B., Kirkup, D., Casanoves, F., Joswig, J.S., Günther, A., Falczuk, V., Rüger, N., Mahecha, M.D. & Gorné, L.D. (2015) The global spectrum of plant form and function. Nature

2. Since the analysis uses a vegetation demography model, one interesting question is how parameters influence ecosystem demography/structure. I like the results showing the sensitivity for different size groups. An additional interesting diagnostic is how the fraction of small/large trees change with parameters. This information can help future modeling practices to diagnose biases in ecosystem structures. In addition, a theoretical analysis by Falster et al. 2018 suggests that the trait influence on growth can change non-linearly with size. It would be interesting to see whether the results of this study are consistent.

Falster, D.S., Duursma, R.A. & FitzJohn, R.G. (2018) How functional traits influence plant growth and shade tolerance across the life cycle. Proceedings of the National Academy of Sciences, 115, E6789–E6798.

3. Of course, allometry can influence the results by a lot. But is a 15% change in the stem allometric coef c (the exponent in the allometric equation) justifiable? This is actually linked to the limitation that no parameter distribution is included. But I would suggest including some discussion for the most sensitive parameters.

4. The most sensitive parameters (e.g. target storage carbon) seems to be a rather model-specific one. What does this imply for other models or ecophysiology?

5. I feel the manuscript can benefit from some re-organization of figures to condense the scientific finding. Most importantly, it seems the sensitivity does not change much with time after a few years, which is expected to me given that only one PFT is included. In this case, I would suggest not to show the changes in sensitivity with time. Instead, just pick two time frame (early succession $\sim$ 5-10 years, and late succession $\sim$ 80-100 years, just like Figure 9) and use bar plots to show how variance is partitioned into different parameters grouped by category shown in Table D1 (Allometry, Photosynthetic, Regrowth, Mortality, etc.).

Stylistic comments: I noticed quite a few typos and inaccurate descriptions over the text. Here I name a few. I would suggest an overall editorial check of the manuscript.

Title missing space between demographically and structured

P1L2 'aimed' to 'that aims'

P3L23 allometry

P7L8 'bare ground', usually it is called near-bare ground since the model assumes a certain seed bank/seedlings to start with.

Figure 9, please make the last panel the same size.

[Figure]

---

## Referee Comment (RC3) · Sebastian Lienert (Referee) · 4 Mar 2019

The presented study performs a sensitivity analysis of parameters in the land model CLM4.5(ED), featuring demographic vegetation. The Fourier Amplitude Sensitivity Test (FAST) is applied to a set of 87 parameters governing vegetation processes. The parameters are sampled using a uniform distribution limited by a +/-15% variation of the default parameter values. The model is run at a single tropical site with one PFT enabled and starting from bare ground conditions. The sensitivities of demographic and carbon cycle quantities are reported and the simulations are qualitatively compared to observations.

The study is successful in identifying key parameters controlling vegetation dynamics in the model in a quantitative manner. This is a useful starting point for further studies optimizing model parameters and investigating parameter related uncertainty. Furthermore, the FAST method is introduced in the context of LSMs. Potential shortcomings of the presented sensitivity analysis are acknowledged in the study, including the choice of the parameter sampling range, potential additional correlation of parameters and the use of a single model configuration at a specific site. Some of the raised caveats might be explored further. For instance, the effect of the used climate forcing could be qualitatively investigated by performing a simulation using climate data of a different gridded reanalysis product (e.g. CRU-NCEP) and comparing it to the observed spread of the simulation ensemble. Nevertheless, I think the study fits the scope of the journal GMD.

In the following some more specific remarks/suggestions:

-Section 2.1: Average period of 30 years: Maybe expand a bit on this choice, would a shorter/longer period substantially alter the results?

-Section 2.4 Data and Model Setup: I think this section is a bit brief and could be improved. The 1x1 degree grid in the first sentence might be confusing since it suggests multiple grid cells. I was also missing information about the atmospheric forcing of $CO_2$ or nitrogen deposition (if enabled).

-Figures 1-6: I wonder if it might not be better to combine the change in the parameters and their respective sensitivities in a single 4x2 figure. This would also reduce the overall number of figures, allowing to include the plot of the number of trees per size class (Figure D2) in the main text, which is quite an important figure in understanding how the sensitivities of the size classes translate to all trees.

-'Most important parameter' in Results 3.1 and Figure 2,4,6 captions: Mention somewhere that this refers to the sensitivity at the end of the simulated period/equilibrium.

-P10L11: Briefly mention again what H2 is (allometric parameters important for vegetation growth)

- Figure 9: Comparison would be easier if rows had identical y-axes. Also, axes are not aligned properly.

Some typos I noticed:

P3L15: Extra space in Farquahr , 1989

P3L18-25: Three times "Therefore, we hypothesize"

P3L23: allmoetry → allometry

P4L10: Missing whitespace: structure.CLM4.5

P9L10: purpose → purposes

P10L1: Extra whitespace after medium

---

## Author Comment (AC1) · 28 May 2019

Dear Editor Müller,

Thank you very much for your time spent handling our manuscript entitled "Identification of key parameters controlling demographically structured vegetation dynamics in a Land Surface Model [CLM4.5(ED)]". The reviewer comments were really useful and we have carefully revised the manuscript to address all of them. Please see below our response to the review comments line by line and the revised manuscript as a supplement. We feel the manuscript is substantially improved and hope that it is now acceptable for publication at GMD.

Yours

Chonggang

On behalf of all coauthors
* * *
Comment: This paper uses the Fourier Amplitude Sensitivity Test (FAST) method to perform a parameter sensitivity study for the CLM4.5(ED) model. The study simulates the variance range and first and second order sensitivities for particular diagnostics relative to parameter perturbations drawn from uniform sampling within $\pm 15\%$ of the model default values for 87 parameters, including biophysical (including temperate response), allometric, allocation, reproduction and establishment, mortality, leaf optical, leaf longevity parameters, and a "competitive exclusion" parameter. The study is performed for one PFT at an Amazon forest site with 25 years of recycled meteorological forcing at 1âŮęx1âŮę, with 5000 simulations (a little less than the âĹij100 x number of parameters called for by Xu and Gertner, 2011, so it seems âĹij37 of the 87 parameters were not "important") 130 years in length.

Response: It is quite rare of have >50 important parameters and thus we feel that 5000 simulations should be adequate for our simulations.

Comment: This study serves more as a template and foundation for further work to perform later parameter optimization and more rigorous analysis relative to data, so as such it provides good documentation of methods of setup and analysis. The authors

largely acknowledge typical concerns about the shortcomings of sensitivity studies like this, including that the parameter sampling is not based on observed distributions, only one PFT is simulated, and the mortality sensitivity to hydraulic failure is not a model sensitivity but rather a site meteorological forcing result. The choice of this particular site was obviously just expedient based on available drivers, so this reviewer views the study more as preliminary setup and test of concept, rather than new findings about DGVMs or about nature. While the results are largely confirming what is already known about the model, such as the overshading of understory trees with the PPA, and the non-linearity of responses by tree size (and some of the results could have been obtained analytically from the equations in the DGVM), the technique illustrates a method quantitatively to stratify or rank the sensitivity of a diagnostic by parameters in a way not available through just Monte Carlo sampling.

Response: We agree with your assessment of our paper. We do want to highlight that this paper represent the first comprehensive sensitivity analysis for FATES and thus should not only provide a framework of sensitivity analysis but also a baseline to understand the model behaviors through time during continued model development.

Comment: For the sake of a sensitivity analysis method, the authors should add more to the discussion about the following:

1) Whether the 25-year periodicity of the meteorological forcings (very apparent in Figures 3c and 3d) affects the purported parameter sensitivity.

Response: Yes, CLM4.5 is sensitive to climate forcing. Following your suggestions, we have added a new figure to show the cycles of climate driver (Fig. 1 in the revised manuscript) and have calculated a rank correlation coefficient between parameter sensitivity indices and the climate driver of temperature, precipitation and relative humidity (Fig. 7 in the revised manuscript). We do found some strong correlations and added one paragraph as following into section 3.2 (Page 14 Line 11-19) as follows,

"...To understand how climate will impact sensitivity results, we also calculated the

Spearman's rank correlation coefficients between the first-order sensitivity index and the corresponding climate drivers (Fig. 10). Our results show that target carbon storage and maintenance respiration rate are negatively correlated with annual mean precipitation and relative humidity, but are positively correlated with annual mean air temperature. This suggests that they are more important for the stressed conditions with low precipitation, low humidity and high temperature. For the leaf allometry coefficient b, it is positively correlated with annual mean precipitation and relative humidity. This suggests the leaf carbon allocation is more important under the favorable environmental conditions for growth with less mortality. In general, our results suggest the climate has a larger impact on the parametric sensitivities for short-term carbon fluxes (GPP and NPP) and vegetation status (LAI) but has a smaller impact on parametric sensitivities for long-term vegetation carbon stocks. . ."

Comment: 2) A little more explanation about the 30-year intervals chosen to average the sensitivity values. The authors say, "This is done in view that the transient and abrupt changes across different size categories in the annual model outputs could make the FAST analysis only account for a minor amount of the variance contribution from each parameter." If the FAST sensitivities are a function of temporal averaging period, this is rather important to address! In that case, it seems running means and a spectral averaging approach would make sense to identify time scales of sensitivity. This perhaps is a missed opportunity to show something interesting in terms of the model sensitivity; e.g. there are model-dependent fast and slow processes updated at short time scales, some processes updated at longer time scales, as well as event–driven processes dependent on meteorological forcing.

Response: Thank you for this comment and it is a great point. We do agree that fast and slow process could play a potential role, which is captured in the manuscript (e.g., GPP and NPP for flow process and accumulation of biomass for a slow process). However, in this case, we think it is mainly because the abrupt transition across different sizes based on the size classification of small, medium and large trees. Thus, we feel

there is no need to do the time-averaging as it will yield very similar results to our current analysis (see new Fig. D1).

Comment: 3) For the sake of readers not familiar with FAST, how does the underlying distribution of parameters propagate to the calculation of the variance, the latter by definition assuming Gaussian distributions. Some sentences on this would be good to add, citing statistical theory papers, and not just application papers.

Response: Thank you for this point. FAST can applicable for any kind of parameter distributions. We have updated the FATES description to capture this (Page 5 line 27-28).We have updated the FAST description section to add theoretical citations for FAST including the theoretical derivation of FAST for parameter interactions (Xu and Gertner 2011a) and the theoretical estimation of sensitivity standard errors (Xu and Gertner 2011b).

Comment: 4) That the sensitivity values change with time and community structure, but the parameters do not change their relative rank to each other: is this a result of the model structure (one PFT, fixed sensitivities in the physics) or a result of the site meteorology?

Response: Thanks for this question. We would like to point out that the rank of parameter sensitivities can change over time, and this can be seen in the sensitivity figures. What we report are the rank of the average parameter sensitivity over the 130-year simulations. So in other words, if a parameter is ranked first in sensitivity, that is because it has the highest mean sensitivity over the simulation period. If a parameter is ranked second in sensitivity, that is because it has the second highest mean sensitivity over the simulation period, etc. The parameter ranking can change through time (e.g., Fig 3 e) and we do add a new figure (Fig 7) to show how the parameter sensitivity is impacted by the climate drivers.

Comment: 5) A suggestion/question: Is it possible to do surface analysis of two-variable sensitivities, and would that reveal any useful relations?

Response: The surface analysis of two-variable sensitivity should be close to the two parameter interactions, which were investigated and shown in the appendices of our manuscript (Figures D6-8, D12).

Comment: Other small things to fix: Section 1 of the paper needs to be proofread for grammar and subject verb agreement. There are a lot of sentences that are a bit sloppy. After the Section 1, this problem disappears!

Response: We have proofread section 1 and it is much improved for grammar.

Comment: Overall, the paper is clearly written, well documented, figures illustrate results well. As the Holm et al. (2018, In Review) paper cited picks up where this paper leaves off with data on the parameter distributions to provide for some quantitative understanding of the system being profiled, this paper by Massoud et al as primarily a methodological exercise is fine for a journal like GMD.

Response: Thank you so much for the positive feedback and helpful comments.
* * *
Comment: This paper applies the Fourier Amplitude Sensitivity Test (FAST) to a land surface model that accounts for vegetation demography. In particular, the authors try to conduct a comprehensive assessment of the sensitivity in the simulated vegetation dynamics to over 80 input parameters that describe plant biochemical, allometric, and demographic traits. The analysis is performed for a tropical rainforest region in the Amazon, where model bears large uncertainty. Limited by data and computational

cost, the paper only included one plant functional type and found that model results are very sensitive to allometric parameters across all time-scales.

Generally, I feel the study is somewhat interesting for GMD in the sense that it introduces FAST to vegetation demography and ecosystem modeling, and that it includes allometric parameters. However, I think the paper can be improved in several aspects to be more useful to the community.

Response: Thank you very much for the positive assessment of our manuscript.

Comment: 1. [P6L32-33] I understand the challenge to include trait-covariation in such analysis. However, the current assumption of absolute orthogonality between parameters makes it hard to interpret the results. For instance, Diaz et al. 2015 shows that the actual ecophysiologically-viable trait space might only be 2% of the total N-dimensional parameter space. It would be helpful to include some more discussions to help interpret the results

Díaz, S., Kattge, J., Cornelissen, J.H.C., Wright, I.J., Lavorel, S., Dray, S., Reu, B., Kleyer, M., Wirth, C., Colin Prentice, I., Garnier, E., Bönisch, G., Westoby, M., Poorter, H., Reich, P.B., Moles, A.T., Dickie, J., Gillison, A.N., Zanne, A.E., Chave, J., Joseph Wright, S., Sheremet'ev, S.N., Jactel, H., Baraloto, C., Cerabolini, B., Pierce, S., Shipley, B., Kirkup, D., Casanoves, F., Joswig, J.S., Günther, A., Falczuk, V., Rüger, N., Mahecha, M.D. & Gorné, L.D. (2015) The global spectrum of plant form and function. Nature

Response: Thank you very much for the great comments. We added a new section in the introduction to distinguish two types of studies: sensitivity analysis study and uncertainty quantification study (page 1, line 30 to page 2, line 8) . Our study is the sensitivity analysis study and thus we did not explore the trait coordination and trade-offs. We have incorporated your suggested paper and points in the discussion section 4.3 for future uncertainty analysis studies.

[Figure]

Comment: 2. Since the analysis uses a vegetation demography model, one interesting question is how parameters influence ecosystem demography/structure. I like the results showing the sensitivity for different size groups. An additional interesting diagnostic is how the fraction of small/large trees change with parameters. This information can help future modeling practices to diagnose biases in ecosystem structures. In addition, a theoretical analysis by Falster et al. 2018 suggests that the trait influence on growth can change non-linearly with size. It would be interesting to see whether the results of this study are consistent.

Falster, D.S., Duursma, R.A. & FitzJohn, R.G. (2018) How functional traits influence plant growth and shade tolerance across the life cycle. Proceedings of the National Academy of Sciences, 115, E6789–E6798.

Response: Thanks for this suggestion and we have now added the suggested sensitivity analysis for fraction of small/medium/large size trees (see Fig D6). Our analysis does support that the trait influence on growth changes with size (e.g., Fig. 5) and we have incorporated your suggested paper and points in our citation in the discussion section 4.1 (see Page 16 Line 1-2).

Comment: 3. Of course, allometry can influence the results by a lot. But is a 15% change in the stem allometric coef c (the exponent in the allometric equation) justifiable? This is actually linked to the limitation that no parameter distribution is included. But I would suggest including some discussion for the most sensitive parameters.

Response: We added to the introduction the distinction between two types of studies: sensitivity analysis study and uncertainty quantification study (Page 2 Line 30-Page 3 Line 8). This paper is a sensitivity analysis study focusing on the understanding of model behaviors. We point out in discussion section 4.3 that ongoing and future studies of uncertainty analysis that specify the parameter distributions will help us understand the uncertainty contributions from each parameter. In the revised manuscript, we do point out why allometrics are important due to their non-linearity and there is a

large amount of variability for allometry coefficients in the reported literature (e.g., Feld-pausch et al. 2011) (see Page 14 line 25-line 35).

Comment: 4. The most sensitive parameters (e.g. target storage carbon) seems to be a rather model-specific one. What does this imply for other models or ecophysiology?

Response: We did point out a lot of similarity to other model sensitivity analysis in section 4.1. In the revised manuscript, we added our comparison to a new sensitivity analysis based on the size-structured model, 3D-CMCC-CNR, by changing the parameters one-at-a-time for 10% deviation from their default values [Collalti et al., 2019]. Their study also showed the importance of allometric parameters. The importance of target carbon storage could be model specific; however, it is in agreement with the main control of plant mortality on vegetation stocks from other models (Sargsyan et al., 2014). Following your suggestion, we also discussed the potential bias of target carbon storage as an important parameter, as carbon starvation is the main mortality mechanism that kills trees in our simulations (see page 15 line 1-6).

Comment: 5. I feel the manuscript can benefit from some re-organization of figures to condense the scientific finding. Most importantly, it seems the sensitivity does not change much with time after a few years, which is expected to me given that only one PFT is included. In this case, I would suggest not to show the changes in sensitivity with time. Instead, just pick two time frame (early succession âĹij 5-10 years, and late succession âĹij 80-100 years, just like Figure 9) and use bar plots to show how variance is partitioned into different parameters grouped by category shown in Table D1 (Allometry, Photosynthetic, Regrowth, Mortality, etc.).

Response: Thanks for this comments. We feel that pick only two time points will lose some important message on the cycle of parametric sensitivity, especially with the newly added Figure 1 and 7 to check the impact of climate on parametric sensitivity, based on the comment from reviewer #2. Following reviewer Sebastian Lienert, we did reorganize the figures by merging the model output ranges and the parametric

sensitivity figures, which saved space for the manuscript.

Comment: Stylistic comments: I noticed quite a few typos and inaccurate descriptions over the text. Here I name a few. I would suggest an overall editorial check of the manuscript. Response: Thank you for pointing this out and we have carefully proofread the revised manuscript by our coauthors.

Comment: Title missing space between demographically and structured

Response: Done. Thank you.

Comment: P1L2 'aimed' to 'that aims'

Response: Done. Thank you.

Comment: P3L23 allometry

Response: Done. Thank you.

Comment: P7L8 'bare ground', usually it is called near-bare ground since the model assumes a certain seed bank/seedlings to start with.

Response: Done. Thank you.

Comment: Figure 9, please make the last panel the same size.

Response: We have updated this Figure 9, which is now Figure 8 in the revised manuscript.

——————————————————————————
Comment: The presented study performs a sensitivity analysis of parameters in the

land model CLM4.5(ED), featuring demographic vegetation. The Fourier Amplitude Sensitivity Test (FAST) is applied to a set of 87 parameters governing vegetation processes. The parameters are sampled using a uniform distribution limited by a +/-15% variation of the default parameter values. The model is run at a single tropical site with one PFT enabled and starting from bare ground conditions. The sensitivities of demographic and carbon cycle quantities are reported and the simulations are qualitatively compared to observations.

The study is successful in identifying key parameters controlling vegetation dynamics in the model in a quantitative manner. This is a useful starting point for further studies optimizing model parameters and investigating parameter related uncertainty. Furthermore, the FAST method is introduced in the context of LSMs. Potential shortcomings of the presented sensitivity analysis are acknowledged in the study, including the choice of the parameter sampling range, potential additional correlation of parameters and the use of a single model configuration at a specific site. Some of the raised caveats might be explored further. For instance, the effect of the used climate forcing could be qualitatively investigated by performing a simulation using climate data of a different gridded reanalysis product (e.g. CRU-NCEP) and comparing it to the observed spread of the simulation ensemble. Nevertheless, I think the study fits the scope of the journal GMD.

Response: Thank you very much for your positive assessment of our manuscript.

Comment: In the following some more specific remarks/suggestions:

-Section 2.1: Average period of 30 years: Maybe expand a bit on this choice, would a shorter/longer period substantially alter the results?

Response: We have explored 20- and 40- year intervals and we get similar results. See the new Figure D1 in the revised manuscript and we have incorporate this point in the manuscript (Page 5 Line 15-16).

[Figure]

Comment: Section 2.4 Data and Model Setup: I think this section is a bit brief and could be improved. The 1x1 degree grid in the first sentence might be confusing since it suggests multiple grid cells. I was also missing information about the atmospheric forcing of CO2 or nitrogen deposition (if enabled).

Response: We have added the climate drivers (Fig. 1) and updated CO2, nitrogen deposition and fire component in the Data and Model Setup section.

Comment: Figures 1-6: I wonder if it might not be better to combine the change in the parameters and their respective sensitivities in a single 4x2 figure. This would also reduce the overall number of figures, allowing to include the plot of the number of trees per size class (Figure D2) in the main text, which is quite an important figure in understanding how the sensitivities of the size classes translate to all trees.

Response: Thank you for this suggestion. We have reorganized all figures of sensitivity results with their corresponding model outputs. Following your suggestion, we have also moved Fig. D2 into the main text as the new Fig. 2.

Comment: 'Most important parameter' in Results 3.1 and Figure 2,4,6 captions: Mention somewhere that this refers to the sensitivity at the end of the simulated period/equilibrium.

Response: Thanks for this good point. What we report are the rank of the average parameter sensitivity over the 130-year simulations. We have pointed out this in Fig 2 caption with other figures refer to that.

Comment: P10L11: Briefly mention again what H2 is (allometric parameters important for vegetation growth)

Response: Done. Thank you.

Comment: Figure 9: Comparison would be easier if rows had identical y-axes. Also, axes are not aligned properly.

Response: We have updated this figure to follow your suggestions on both x-axes and y-axes.

Comment: Some typos I noticed:

P3L15: Extra space in Farquahr , 1989

Response: Done. Thank you.

Comment: P3L18-25: Three times "Therefore, we hypothesize"

Response: We have revised the manuscript to avoid the repetition.

Comment: P3L23: allmoetry → allometry

Response: Done. Thank you.

Comment: P4L10: Missing whitespace: structure.CLM4.5

Response: Done. Thank you.

Comment: P9L10: purpose → purposes

Response: Done. Thank you.

Comment: P10L1: Extra whitespace after medium

Response: Done. Thank you.
* * *
Response to short comment by Vick Prasad vik7prasad@gmail.com

Comment: In this study, the authors used a type of sensitivity analyses method to understand the behavior of a developed land-surface model (CLM4.5(ED)) to changes in parameter values. They arrive at results which other land surface models such as those that focus more on bio-geochemistry (e.g. CLM4.5) or those that focus more on

vegetation dynamics (e.g. ED) could have also come up with if these models were ran separately. While their sensitivity analyses method is okay, their results are trivial. I have concerns regarding the actual simulation method they have used in this study.

Response: This is the first comprehensive sensitivity analysis for FATES. In this study, we considered > 80 parameters and assessed their sensitivities for carbon fluxes, stocks, growth and mortality for different sizes of trees and their linkage to climate conditions. In view that FATES is the next generation dynamic vegetation model with a large user community that supported by both DOE and NCAR, our analysis on FATES should guide the understanding of model behaviors and thus help future model parameterization, improvement and applications. It is a very important step for FATES and will also be potentially helpful for other models with a similar data structure.

Comment: My concerns are on the bias and uncertainties the authors may have in their results. I list the points below, which are likely to be interconnected.

1) The entire parameter space of the model was not explored so how can this be even called a global sensitivity analyses? You looked at 87 parameters in this study. How much uncertainty you have in your existing results for the parameters that you have ignored?

Response: There are two types of uncertainty and sensitivity studies for models. One type of study aims to understand the model behaviors by exploring the baseline sensitivity of model outputs to parameter changes, which is normally an equal amount of deviation from the mean values of default parameters. They are commonly referred to as model sensitivity or elasticity analysis studies [e.g., Benton and Grant, 1999; Collalti et al., 2019; Menberg et al., 2016] . Another type of study aims to quantify how much uncertainty is in the model outputs and what contributes to the uncertainty. They are commonly referred to as uncertainty quantification studies. It is possible that a parameter is very sensitive for a model output in the sensitivity analysis study, but could contribute to a small amount of uncertainty in the model output in the uncertainty quantification study there is only a small amount of uncertainty in the parameter estimate. Both types of studies are useful for model development with the sensitivity analysis studies focusing on understanding the baseline of model behaviors and the uncertainty quantification studies focusing on guiding field and laboratory measurements [e.g., Xu et al., 2010]. Our study belongs to the sensitivity analysis studies. The main difference between these two types of studies lies in the difference in research goals with similar uncertainty and sensitivity analysis approaches. We have laid out these two types of studies in the introduction (page 1, line 30 to page 2, line 8) to clarify the purpose of this paper.

The "global" in this paper refer to the fact that we change all the parameters simultaneously for understanding the impact of parameter of model outputs given the parameter range we defined. There is confusion on how we define "global" sensitivity analysis. A sensitivity analysis is considered to be global when all the input factors are varied simultaneously and the sensitivity is evaluated over their entire range of interest [McRae et al., 1982; Xu and Gertner, 2008; Zhou et al., 2008]. From methodological perspective, the global sensitivity analysis changes the parameters simultaneously. From the scientific perspective, the global sensitivity analysis samples the entire range of interest. For sensitivity analysis studies, the entire range of interest could be a certain percentage of default values of parameters. For uncertainty quantification studies, the entire range of interest could be the distributions of parameter estimated from laboratory measurements, field observations and expert knowledge. Campolongo [2000] suggested to classify local and global sensitivity analysis based largely on the extent of the input variable range that the technique assesses; however, this arrangement is ambiguous because the classification depends on whether the range is sufficiently large to be perceived as global [Song et al., 2015]. We have clearly laid out these details in our introduction to avoid confusion (page 3, Line 17-24).

Comment: 2) How does these results of CLM4.5(ED) compare with the other versions of CLM e.g. with CLM-DGVM or CLM-FATES?

Response: This version of model [CLM4.5(ED)] is the initial version of CLM-FATES. We changed the name from ED to FATES about 2 years ago. We have made the name change from ED to FATES to avoid confusions. As far as we know, we are not aware of studies for CLM-CNDV, which is an original version of the dynamic vegetation for CLM. We did compare our results with CLM and ED in the manuscript (Section 4.1). We also added our comparison to a new sensitivity analysis based on the size-structured model, 3D-CMCC-CNR, by changing the parameters one-at-a-time for 10% deviation from their default values [Collalti et al., 2019]. Their study also showed the importance of allometric parameters. We have incorporated this comparison in the discussion section 4.1.

Comment: 3) The model simulations are performed for 1 deg x 1 deg (approximately at 100 km). This resolution is quite coarse. If you are trying to understand the large-scale vegetation responses to changes in parameter values, then I think that needs to be made clear (at-least in the abstract as well as in the introduction). If not, then you need to address how much your results will change if you did the sensitivity analyses at the local scale using local weather conditions.

Response: We have included a statement on the resolution for both abstract and the introduction. We also discussed the model sensitivity to climate drivers in section 4.3 (Page 21 Line 16-26).

Comment: 4) This simulation is only carried out at one site. Why was this specific site chosen? Isn't this already a bias? Will you get similar results at other biomes?

Response: This a good question. We chose this site because CLM (FATES) is already set up for this site and it is common test site for the tropical biome. We will expect to see different results for other biomes, or possibly other tropical sites with different climates, but we will expect the main results will stay valid. That is, what parameters fell out as driving the model uncertainty would still show up for a different tropical site. This is the first sensitivity test of CLM (FATES) and we know other research groups

are working on different sites to improve our understanding of the model at different locations. We have included a statement in the section of 4.3 as below:

" ... it is possible that the parameter sensitivity could be different if we use different model inputs, different sites, and different structures of subcomponents within the model. For example, using site level climate drivers, instead of the reanalysis meteorological drivers used in this study could lead to different sensitivity values since our preliminary analysis showed that simulated vegetation demography is quite sensitive to different climate drivers. Furthermore, there are ongoing development activities to improve different components of the models. For example, there are current efforts to incorporate different representations of tree allometry within CLM4.5(FATES), which have different formulations between size and biomass, the allocation of nitrogen and thus the photosynthetic process. Therefore, model improvements such as these can affect corresponding sensitivity analysis results. To understand the impact of site level variations on model dynamics, similar sensitivity analysis across different sites can be conducted to understand how climate variability will affect the sensitivity analysis results. "

Comment: 5) The climate data was recycled, which might be okay, but you used climate data from 1942 to 1972? I don't think you can compare your modeled results with observations unless you believe that the climate at your studied site didn't change much or if your measurements were carried out around 1972? Further, isn't CLM4.5(ED) sensitive to climate forcing?

Response: Yes, CLM4.5 is sensitive to climate forcing and we incorporate a new paragraph to assess the climate condition on parametric sensitivity (Page 13 Line 32-Page 14 Line 6). We understand your concern about the climate driver and I agree with you of the potential bias in model-data comparison. We want to point out that it is common for CLM simulations to use recycled climate and in the paper that comparison with data is meant as a range/sanity check but not a validation, in view that main focus of the paper is on the relative impact of different parameters on model output. We do point out

this potential bias in our comparison in section 4.1 (page 16, line 28-30).

Comment: 6) The simulation was carried out for about 130 years, where the changes in parameter values (+/- 15%) was relatively small compared to the default value. This % change was fixed for all parameters. Isn't there any parameter out of 87 that has a wider range in reality. If so, how can one be really sure about these results then?

Response: As we pointed out at beginning of response, this study is a sensitivity analysis study focusing on understanding of the model behaviors in our response to your comment #1. Ongoing studies of uncertainty quantification will help us understand the uncertainty contributions from realistic parametric distributions. We have also laid out this caveat in our section of limitation of methods in the discussion (section 4.3).

Comment: 7) The authors should quantify the relative impacts on the carbon fluxes or vegetation stocks due to parameter changes, and state whether these impacts are statistically significant or not. At present, it is unclear how much the identified parameters control the carbon fluxes or stocks.

Response: We would like to point the reviewer to Fig. 1-6 where we did plot the sensitivity indices of parameters (i.e., the proportion of variance in the model output contributed by each parameter, which measures the relative impacts of parameters on carbon flux and vegetation stocks) through time for carbon fluxes and stocks. We also plotted the impact of key parameter on GPP, NPP, LAI and Biomass through cubic splines (see Fig. 8 in the revised manuscript). Thus, we are a bit confused regarding the suggested change by the reviewer. Do you want us to plot the contributions of identified important parameters to fluxes or vegetation stocks on a single plot for an easier comparison of the relative contributions? We do have the standard errors of estimated sensitivities based on the delta method. See details from Chonggang Xu & George Gertner (2011) Reliability of global sensitivity indices, Journal of Statistical Computation and Simulation, 81:12, 1939-1969. Following our understanding of your suggestion, we have introduced the standard error estimation in FAST description

(Page 6 Line 26-28) and plotted the sensitivity and associated standard errors in the updated Fig. D9.

Please also note the supplement to this comment:
https://www.geosci-model-dev-discuss.net/gmd-2019-6/gmd-2019-6-AC1-supplement.pdf

**Supplement:**

[revised manuscript text omitted]

---

## Author Response (AR2)

Dear Editor Müller,

    Thank you very much for your time spent handling our manuscript entitled "Identification of key parameters controlling demographically structured vegetation dynamics in a Land Surface Model [CLM4.5(ED)]". We have addressed the minor revision suggestions from review #2 and hope that it is now acceptable for publication at GMD.

Yours

Chonggang Xu

On behalf of all coauthors

**Responses to Referee #2 (Xiangtao Xu, xu.withoutwax@gmail.com)**

**Comment 1:** I am generally happy with the revision. It helps to improve the manuscript by a lot. The correlation analysis between climate forcing and sensitivity indices are helpful and make sense. The re-organization of figures improves readability. My only suggestion is to connect such kind of computational sensitivity analysis with theoretical ecological analysis in vegetation demography in the Conclusion part. The FAST method presented here can provide a more comprehensive analysis of the parameters that control vegetation demography while simpler but more tractable theoretical ecology models (e.g. Farrior et al. 2016 and Falster et al. 2018) can provide more mechanistic understanding (e.g. why and how allometry influences demography). Such modeling exercises at different scales and complexity are complementary, especially when vegetation models are getting more and more complex nowadays.

Farrior, C.E., Bohlman, S.A., Hubbell, S. & Pacala, S.W. (2016) Dominance of the suppressed: Power-law size structure in tropical forests. Science, 351, 155–157. Falster, D.S., Duursma, R.A. & FitzJohn, R.G. (2018) How functional traits influence plant growth and shade tolerance across the life cycle. Proceedings of the National Academy of Sciences, 115, E6789–E6798.

**Response 1:** This is an excellent point and per your suggestion, we have added the following sentence to the end of discussion section of the manuscript:

Although the FAST method presented in this study can provide a more comprehensive analysis of the parameters that control vegetation demography, it is mostly built on statistical relationships (e.g., Fig.8). A complementary approach is to use more tractable theoretical ecology models (e.g. Farrior et al. 2016 and Falster et al. 2018) to approximate the underlying model input-output relationships, which can provide more mechanistic understanding of model behavior.

**Comment 2**: In addition, Page 19 Line 1-5 discussed the model-data comparison for mortality. My understanding is that the authors extracted and averaged instantaneous mortality in the model, which is calculated differently from the census-based mortality. For example, if all the trees of a given size class died within a 10-year census, the mortality from the observation would be 0.1 per year although they might have already died in the first few years (i.e. actual mortality is higher). So I am not so surprised at the modeled mortality for small trees is higher than the

observed because they are at different time scales. An apple-to-apple comparison should simulate census processes in the model as well, which involves tracking cohorts in the model. This is definitely too much for this manuscript but it is worthy of mentioning in the discussion to avoid confusion.

**Response 2**: Thank you for pointing this out and we have added the following sentence to the manuscript,

"It is also possible that the observed mortality rate for small trees could potentially be underestimated if all the trees in a certain size classes died at a shorter time frame than the census intervals."